

# V2I-VTL: IoT-Enabled adaptive traffic light controller and emission reduction at intersection

Ajmal Khan[1], Shams ur Rahman[2], Farman Ullah[3], Muhammad Ilyas Khattak[4], Mohammed M. Bait-Suwailam[1] and Hesham El Sayed[3]

[1] Communication and Information Research Center, Sultan Qaboos University, Muscat, Oman
[2] Department of Computer Science, University of Engineering and Technology Mardan, Mardan, Pakistan
[3] Computer and Network Engineering Department, College of Information Technology, United Arab Emirates University, Al Ain, United Arab Emirates
[4] School of Control Science and Engineering, Shandong University, Shandong, China

## ABSTRACT

Traffic congestion is a growing concern in urban centers worldwide, leading to significant delays, particularly for emergency vehicles such as fire trucks and ambulances. This not only increases emergency response time but also the risk of life and property loss. To address this issue, our research introduces a traffic control system that prioritizes emergency vehicle egress and mitigates intersection congestion. Other than validation through simulation, the system's efficacy is further substantiated by real-world hardware implementation. The system employs an access point (AP) at intersections to receive location and direction data from approaching vehicles. Emergency vehicles are given precedence, while non- emergency vehicle data is used to adjust traffic light durations, thereby optimizing the traffic flow. The simulation results demonstrate the system's reduced lane opening times and average waiting periods for emergency vehicles. Advancing from simulation to application, we have executed a real-world hardware validation at a high-traffic intersection. This phase entailed the precise installation and calibration of the necessary hardware components, transitioning from theoretical models to practical, operational technology. The hardware setup confirms the system's practical viability and offers a more comprehensive assessment of its impact on traffic efficiency and emergency response times. This dual approach of simulation and hardware validation provides a thorough evaluation of the system's capabilities, establishing a foundation for future traffic management solutions. Additionally, the implementation of the system leads to a notable reduction in $CO_2$ emissions at intersections, contributing to environmental sustainability efforts.

# INTRODUCTION

In recent years, many cities all over the world have encountered a rapid population increase due to a rising desire to live in urban regions (*Kuddus, Tynan & McBryde, 2020*). This has led to a rapid increase in vehicle traffic volume. Therefore, more accidents, longer emergency response times, and traffic congestion have occurred as a result of the increased

Corresponding author
Mohammed M. Bait-Suwailam, msuwailem@squ.edu.om

traffic volume in urban areas (*Afrin & Yodo, 2020*). The use of vehicles such as cars and vans has increased dramatically as a result of urbanisation around the world. For example, between 1990 and 2019, the total number of automobiles in the United States climbed by 60 percent (*Statista, 2022*). Moreover, more than 50 percent of the combined total of fatal and injury crashes occur at or near intersections as mentioned in the Federal Highway Administration (FHWA's) report (*U. S. Federal Highway Association, 2021*).

Long lines, traffic bottlenecks, and hampered emergency response are all consequences of high traffic volume. In the event of an emergency, emergency vehicles (EVs) must respond swiftly and arrive at the emergency location as quickly as possible, allowing first responders to commence search, rescue, and first-aid operations. Any delay in the arrival of EVs can lead to further loss of life and property. As an example, about 700 fatalities were recorded in Ireland annually due to hindrance and late response of the ambulances (*Djahel et al., Dec 2015*). The increased vehicle density at intersections not only increases the delay of rescue operations but also increase the number of EV collisions. This is because high speed EVs can run into others vehicles after entering an intersection and thus increases the chances of accidents. The National Highway Traffic Safety Administration (NHTSA) reported that about 4500 motor vehicles were involved in annual traffic accidents including 1500 ambulance crashes from 1992 to 2012 (*Blincoe, June 6, 2021*). According to the National Safety Council (NSC) (*Watanabe et al., 2019*) report, fire engine accidents were the second most common cause of firefighter injury. Over the ten-year period, there were over 31,600 fire truck accidents. Furthermore, police chasing causes about 300 deaths in the United States each year.

In addition to causing accidents, traffic congestion at intersections significantly contributes to environmental pollution. Traffic congestion depends on many factors, including environmental conditions such as rain and snow, as well as the time of day. For example, in urban areas, traffic at intersections is lighter during weekends and heavier during weekdays (*Minglei, Rongrong & Binghua, 2020*). Traffic congestion is one of the leading causes of various forms of environmental pollution, particularly air pollution. Smog is caused by the emission of carbon dioxide and sulfur dioxide from automotive engines. This pollution severely contaminates the air, contributing to global warming while also obstructing the vision of drivers, passengers, and pedestrians (*Bhatlu et al., 2020*).

Traditional traffic lights are commonly used to manage traffic congestion at intersections. Signals at intersections are turned on and off for predetermined time periods in traditional traffic management systems, without taking into consideration the real traffic flow at junctions. Moreover, the conventional traffic control systems are unable to prioritize EVs, such as, ambulances, fire brigade, *etc*. In such a traffic control system, vehicles must wait for green light to turn on before crossing the intersections. Moreover, even if there are no other vehicles at intersection, the fixed time traffic controller will assign it a green signal, thereby increasing the waiting time of vehicles on other approaches. Hence, it is evident that delayed emergency response can lead to loss of human lives and property. This delay can be significantly reduced with use of a proper traffic lights control system. However, as explained earlier, fixed time traffic controller are unable to assign priority to EVs. As a

result, developing an effective and intelligent traffic control system that can perform well in both regular and emergency traffic scenarios is essential.

To minimize urban traffic congestion, previous studies have proposed various algorithms to reduce vehicular delay at intersections like smart vehicular systems (*Li et al., 2020*). Smart vehicular systems have solved many challenges faced by automated vehicles. Most modern-day vehicles are equipped with advanced technology like cruise control, Global Positioning System (GPS)-based route planning, automatic braking and steering control using sensors and actuators (*Liu, Lu & Shladover, 2019*). Therefore, there is a need for an intelligent intersection control system that prioritizes the transit of emergency vehicles while also reducing the waiting time for regular traffic. Hence, the aim of this research is the development of intersection control system that prioritizes emergency vehicles and reduces routine traffic congestion at intersections by utilizing V2I communication. For this purpose, an access point (AP) will be installed at intersection to manage the smooth traffic flow of emergency and non-emergency vehicles. The AP will monitor beacon messages from all vehicles at intersection and will prioritize approaches based on vehicle density on each approach. An approach with higher vehicle density will be opened first, and then the approach with the second highest density will be opened next. Therefore, the main contributions of this research work are:

–To design and develop a smart vehicular control system which manages traffic congestion
–To deploy an access point (AP) at intersection to smoothly manage the transit of emergency and non-emergency vehicles at intersection.
–To enable vehicles to communicate with AP to share position and direction information.
–To compare the performance of the proposed system with existing schemes in terms of lane opening time and total delay experienced

The rest of this article is structured as follows. The summary of related works is provided in 'Related Work'. 'Research Methodology' outlines the major components and research methods of proposed scheme. The simulation setting and results are described in 'Simulation-Based Performance Evaluation'. The article is concluded in 'Real-World Hardware Validation'.

## RELATED WORK

Emergency vehicle accidents are a severe problem that is becoming more and more prevalent worldwide. The majority of crashes involving emergency vehicles take place close to intersections due to the fact that they move more swiftly in emergency circumstances, which can cause significant damage or death. To efficiently address the issue of traffic congestion in urban areas, facilitate the movement of emergency vehicles, and manage vehicular traffic at intersections, researchers have proposed a number of traffic management plans based on vehicle-to-infrastructure (V2I) and vehicle-to-vehicle (V2V) communication.

In *Belyaev et al. (2015)*, cameras are installed at intersections to provide live traffic surveillance of traffic at intersection to monitor traffic jams and accidents. A VTL based system is proposed in *Tonguz & Zhang (2020)* where each vehicle is equipped with Dedicated Short-Range Communication (DSRC) that transmits vehicle information

to the intersection. Each intersection is equipped with four directional antennas that receive vehicle information from respective directions. *Ghazal et al. (2016)* used a traffic management system to control traffic light using PIC controller to observe traffic density and flow of the dense traffic *via* infrared sensors. The sensors data is then utilized to switch the traffic lights accordingly. Moreover, a portable device is utilized by traffic warden to connect with the traffic management system *via* Zigbee transceiver to facilitate smooth exit of the emergency vehicle at intersection.

*Duarte et al. (2023)* suggested a solution utilizing 6G-enabled VANET-based V2I Communication for the exchange of data between emergency vehicles and traffic lights. This approach aims to improve safety and decrease response times. The SafeSmart 6G system anticipates the arrival time of emergency vehicles at intersections through the analysis of historical data and AI-driven insights. It then requests signal preemption along the selected route to facilitate smoother passage. However, the dependence of the proposed system limits its utilization to regions where 6G exists (or will exist). *He et al. (2023)* introduce a collaboration framework involving vehicle-to-infrastructure (V2I) and vehicle-to-vehicle (V2V) communication, incorporating non-orthogonal multiple access (NOMA) and successive interference cancellation (SIC) techniques. The authors formulate a power consumption minimization problem, considering mode selection, power control, and channel state information latency, and provide a solution through decoupling and graphical methods. The work is focused on minimizing the power consumption rather than emergency vehicle priority.

*Huang, Weng & Zhou (2015)* have proposed a traffic control system which includes two sensors for detecting EV *i.e.,* sensor in and sensor out at intersection to detect arrival and departure of EV respectively. When an EV approaches an intersection, the sensors detect EV and send a message to traffic light controller that shows a green light to respective side. Moreover, when EV leaves the intersection, the sensor out sends a message to the controller to resume its normal operations. In *Doolan & Muntean (2017)*, a V2V concept is suggested, in which vehicles interact with one another to transmit information regarding vehicle density and road conditions, such as average vehicle speed and road surface quality. This data depicts a fuel-efficient model in which cars can choose low-fuel-consumption routes to avoid air pollution and vehicle delays. The main purpose of the proposed scheme was to sense the roughness and irregularities on the roads and to inform neighbour vehicles *via* warning messages. In *Tonguz & Viriyasitavat (2016)*, a vehicle selects a leader vehicle which serves as a traffic controller at intersection to provide traffic information to the remaining vehicles. Vehicles approaching at intersection first elect a cluster leader for each direction *e.g.,* north, south, east and west. One of the cluster vehicle is randomly chosen as a VTL leader.The left approach is given a green light by the VTL leader, whereas the other approaches are given red lights. The VTL is transferred to the cluster leader in its proper direction once a predetermined amount of time has passed. However, the VTL system has a few shortcomings, including a random selection procedure for the leader vehicle that will prolong vehicle wait times at each approach. Moreover, when EV is in the same direction as VTL leader, the leader cannot prioritize EV as the leader vehicle would first handover leading task to vehicles on its left approach.

Recent research explores innovative approaches to adaptive traffic light control using IoT and AI technologies. Deep reinforcement learning has been applied to optimize traffic flow in real-world scenarios, reducing vehicle queue lengths and waiting times compared to fixed-time systems (*Damadam et al., 2022*). Visible Light Communication (VLC) has been proposed for secure vehicle-infrastructure communication, enabling adaptive traffic control through a mesh/cellular hybrid network (*Vieira et al., 2024*). The EVATL framework integrates emergency vehicle communication with adaptive traffic lights, using GPS and IoT to prioritize emergency vehicles while optimizing overall traffic flow (*Dodia et al., 2023*). Deep learning techniques have also been employed to process high-dimensional sensory inputs, such as GPS traces from connected vehicles, eliminating the need for manual feature extraction and outperforming conventional reinforcement learning approaches in minimizing intersection delays (*Shabestary & Abdulhai, 2022*). These advancements demonstrate the potential for intelligent traffic management systems to significantly improve urban mobility.

IoT-based solutions have also contributed to enhancing energy efficiency in smart cities. In *Chen, Sivaparthipan & Muthu (2022)*, IoT-based smart street lighting systems with LED lamps and sensor-controlled dimming have shown potential in significantly reducing energy consumption and carbon emissions. Moreover, integrating VLC with learning-based traffic signal control has enhanced intersection efficiency by enabling vehicle-to-vehicle and infrastructure-to-vehicle communication, reducing waiting times for both vehicles and pedestrians (*Vieira et al., 2024*). An optimization-based framework for autonomous intersection crossing under V2X communication improves traffic control performance, especially in light to medium traffic volumes (*Lu, Jung & Kim, 2022*). The CoTV system, utilizing multi-agent deep reinforcement learning, cooperatively controls both traffic light signals and connected autonomous vehicles to balance the reduction of travel time, fuel consumption, and emissions, while remaining scalable to complex urban scenarios (*Guo, Cheng & Wang, 2022*). These advancements contribute to smarter, more efficient, and sustainable urban transportation systems.

*Hosseinzadeh et al. (2023)* have suggested a traffic management strategy aimed at easing congestion within a network of interconnected lanes and roads with signals. The approach prioritizes emergency vehicles, ensuring they receive swift and efficient routing. The proposed strategy employs model predictive control to regulate incoming traffic flows through network gates and adjust the configuration of traffic lights throughout the network. In *Bui, Camacho & Jung (2017)*, intersection controller and sensors are deployed across the intersections which are then connected to the main cellular base station (CBS). Vehicles upon reaching at intersection request for green signal. The intersection controller checks the respective side and decides whether to give red or green signal on the basis of First In First Out (FIFO) model. Centralized base station connects all intersection controllers with each other and also allows the pedestrian, vehicle or any object having smart phone to connect with intersection controller for exit request through CBS.

*Hu et al. (Dec. 2017)* proposed a multi intersection model (MIM) based on cellular automata (CA) that can tell drivers how much time it would take to go to their destination by staying at different intersections. After gathering traffic density data, the Single Junction

Volume Algorithm (SIVA) calculates the overall volume of cars at the intersection as well as the volume of each lane. The Single Intersection Light Timing Plan Algorithm (SISTPA) will then compute the time for each lane to reach the green signal. In *Chowdhury (2016)*, an Emergency Priority Code System (EPCS) is devised which determines the incident type and priority level and then informs all the traffic controllers about the EV path which enables the traffic controllers to automatically open the respective lane until EV passes *via* intersection. There are also works that have applied neural networks to solve the issue of emergency vehicle prioritization at intersection. For example, in *Feng et al. (2024)*, introduces an Emergency Vehicle Priority Scheduling Model that uses Heterogeneous Feature Fusion in Graph Convolutional Networks to dynamically adjust signal control strategies based on traffic flow predictions. The model ensures prioritized passage for emergency vehicles, reducing congestion in the road network. Experimental results show enhanced traffic flow prediction accuracy, improving overall system efficiency and safety in various scenarios.

Additionally, advancements in machine learning and IoT technologies have been applied to traffic management in smart cities. Adaptive traffic management systems with accident alert capabilities can reduce congestion and improve safety by dynamically adjusting traffic signals based on real-time data (*Balaji et al., 2023*). Edge computing and computer vision techniques, such as YOLO object detection, enable efficient traffic density estimation and signal optimization (*Hazarika et al., 2024*). A parallel emission regulatory framework has been proposed to address vehicle emissions in intelligent transportation systems, incorporating modern aftertreatment systems for more accurate estimations (*Sun et al., 2023*). Furthermore, advanced autonomous intersection management systems utilizing multi-agent deep reinforcement learning have shown significant improvements over traditional traffic light control methods, reducing travel time by 59% and congestion-related time loss by 95% (*Guillen-Perez & Cano, 2022*). These advancements contribute to more efficient and sustainable urban transportation systems.

Existing protocols address the problems of regulating the sequences and duration of traffic light signals in accordance with the vehicle densities on specific road segments or approaches, or they address the problems of prioritising emergency vehicles at junctions. Our earlier research in *Khan et al. (2017)* does not take into account the presence of emergency vehicles at a crossroads and merely adjusts the timings of green traffic signals in line with the detected real-time traffic. The availability of rescue vehicles is analysed using force resistive sensors positioned close to crossings in our next research project (*Khan et al., 2018*). Few methods—namely, giving emergency vehicles priority at junctions and predicting the length of traffic light signals based on vehicle density—address both problems, as far as we are aware. The main contribution of this study is a system that favors emergency vehicles at intersections and controls the order and length of traffic light signals based on the density of automobiles on each approach. This density is assessed by integrating DSRC devices in each vehicle to transmit their positions to access points. The number of cars in each approach is then computed by the access point. The algorithms used by emergency vehicles, non-emergency vehicles, and access points located at intersections are also described in depth. Additionally, thorough simulations are conducted to show

how the suggested approach takes into account the arrival of emergency vehicles in order to shorten their response times at junctions.

## RESEARCH METHODOLOGY

In this research work, we propose a Vehicle-to-Infrastructure (V2I) based Virtual Traffic Light Controller named V2I-VTL that aims to enhance traffic management and reduce emissions at intersections. The system leverages communication between vehicles and infrastructure to optimize traffic flow, prioritize emergency vehicle egress, and mitigate congestion. By dynamically adjusting traffic light durations based on real-time data, our solution contributes to both efficient traffic movement and environmental sustainability

- **Access point:** AP is installed at intersections to control the EV and non-EV vehicles. AP is responsible for the transit of EV on a priority basis and also controls normal traffic congestion.
- **On-board unit:** An on-board unit (OBU) is deployed inside a vehicle to transmit the vehicle's position and direction information to the AP.
- **Emergency vehicle:** An emergency vehicle (EV) approaching an intersection transmits the EV alert message to AP for priority assignment.

### Vehicle to infrastructure virtual traffic light (V2I-VTL)

The proposed V2I-VTL is a traffic control system that immediately reacts to EV arriving at the intersection and handles the smooth flow of non-EVs as well. V2I-VTL consists of two main components *i.e.,* OBUs and APs. An OBU is installed in each vehicle which is responsible for transmitting vehicle's position and direction information to AP. Moreover, OBU is also responsible to receive traffic light green signal information from the AP. Each OBU transmits a beacon message every second that includes vehicle position, direction and velocity information. Similarly, an OBU is installed in EVs like fire brigades, ambulances, police vans, military vehicles etc. to transmit their local information to AP. On the other hand, the AP is positioned at the intersection as shown in Fig. 1. The AP is responsible to collect beacon messages transmitted by EVs and non-EVs. The AP assigns the highest priority to the lane with an EV to cross the intersection. In addition, the AP also prioritizes the regular traffic flow by calculating the traffic density in each direction which is explained later.

#### Vehicle on board unit

The OBU is responsible for two major tasks. First, it transmits a beacon message that includes direction, velocity, and location coordinates of vehicle received *via* GPS. Second, OBU receives the AP's virtual traffic light information to determine whether or not to stop at the intersection. However, two different types of OBUs are proposed in this research work *i.e.,* one for non-EVs and one for EVs. The OBU for non-EV as shown in Fig. 2 contains a controller module which is responsible for communication between user interface and DSRC transceiver. Inside the controller module, there are two sub-modules: the transmitter sub-module which transmits a beacon message containing vehicle ID, beacon ID, velocity, direction and GPS coordinates of the vehicle to the AP through DSRC communication

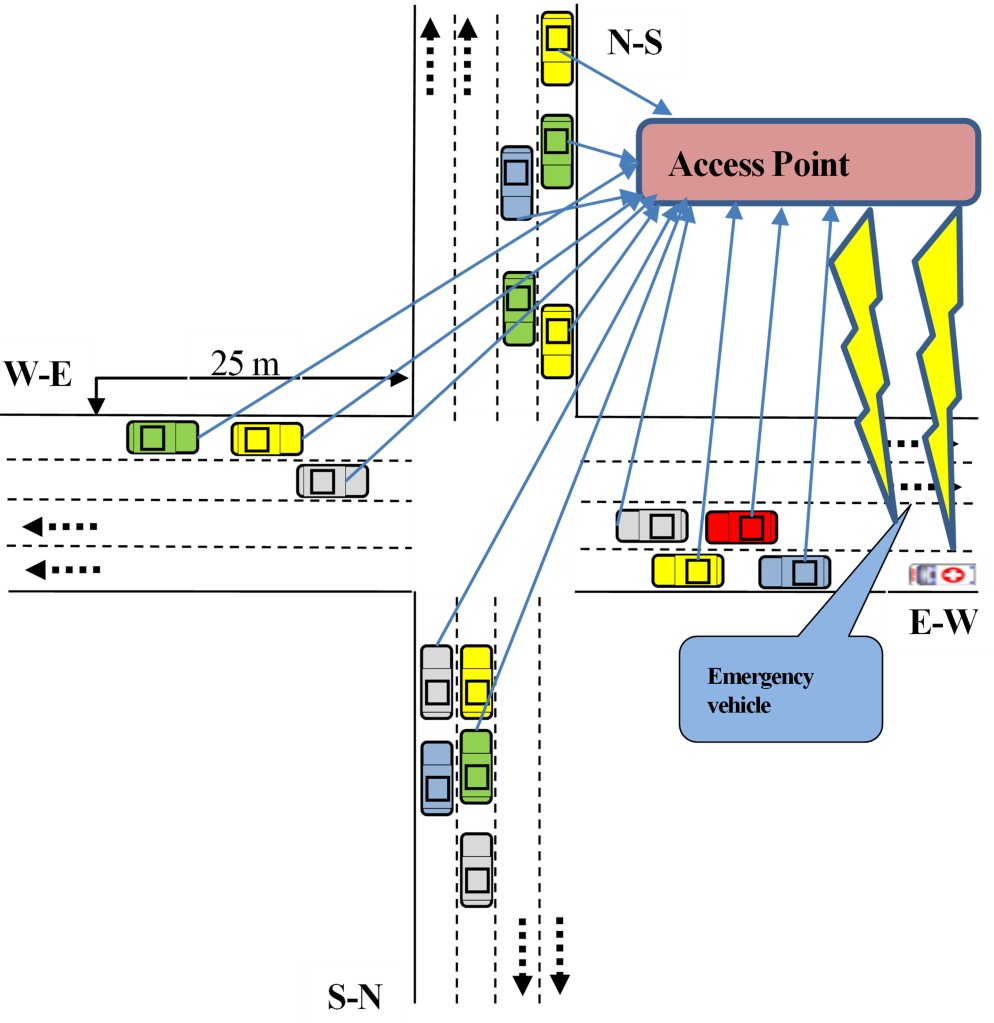

**Figure 1** Access point positioned at the centre of intersection.

and the reception sub-module which receives the VTL information and sends it to the user interface.

Figure 3 shows an OBU for EV in which the transmitter module transmits a beacon message (containing direction information, GPS coordinates of the vehicle, Emergency Priority Request (EPR) and a clear flag) to the AP through DSRC communication. The EPR field is set to 1 by the emergency vehicle to alert the AP regarding the arrival of the EV. The clear flag is set to 1 when the EV crosses the intersection alerting AP to resume its normal operation . The reception module receives the VTL information and sends it to the user interface as shown in Fig. 3. As soon as an EV exits the intersection, it transmits a beacon message having a clear flag set to 1 to allow AP resume normal traffic operation.

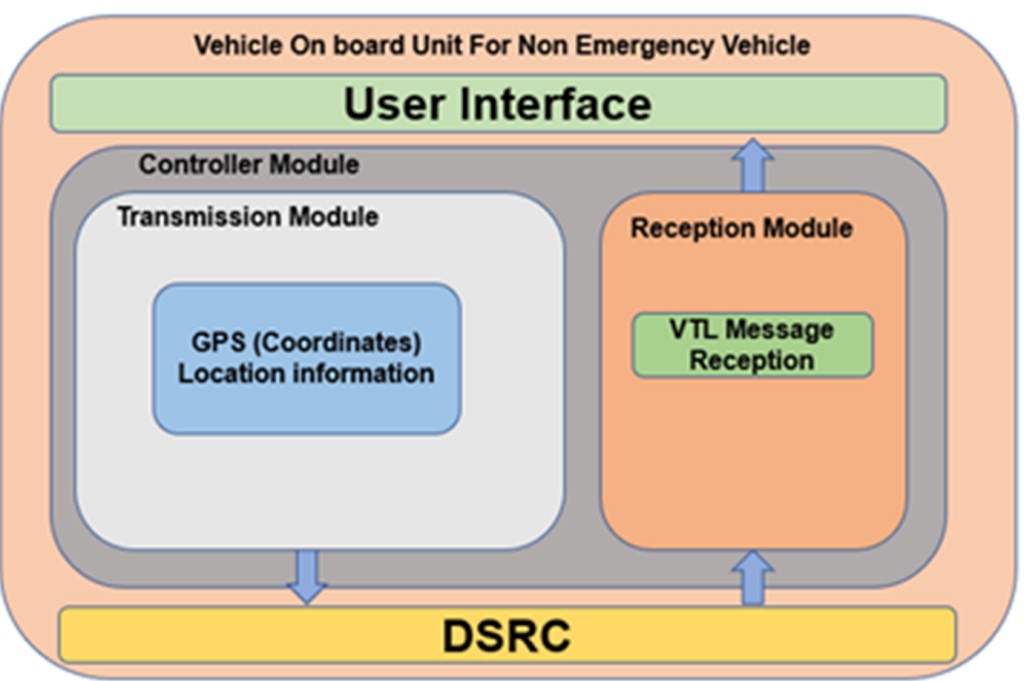

**Figure 2** Block diagram of OBU for non EV.

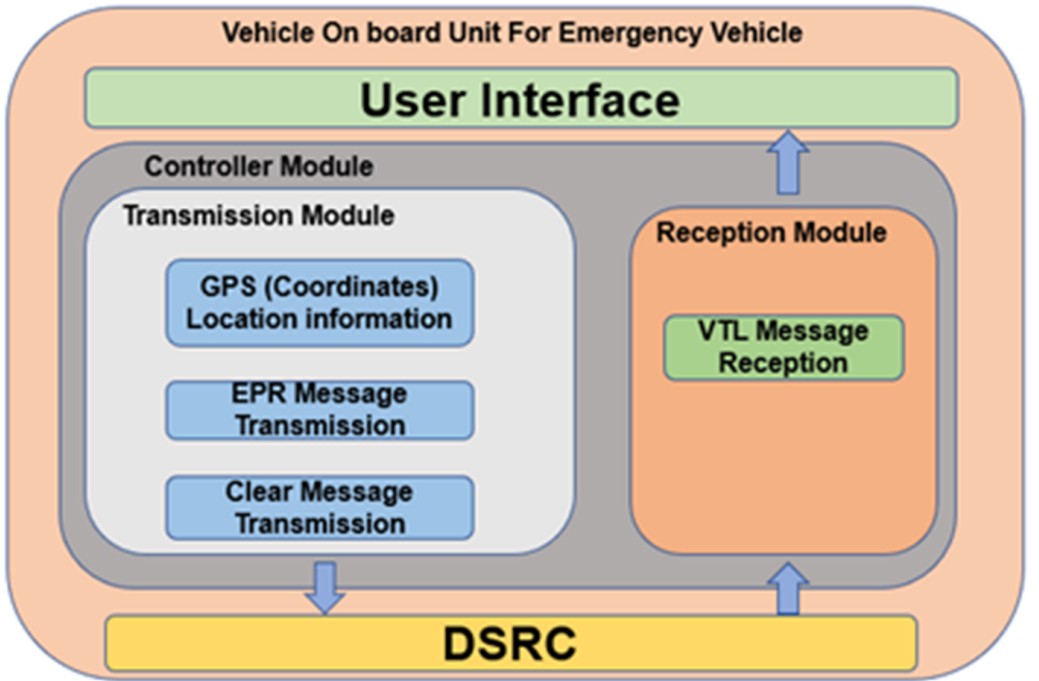

**Figure 3** Block diagram of OBU for EV.

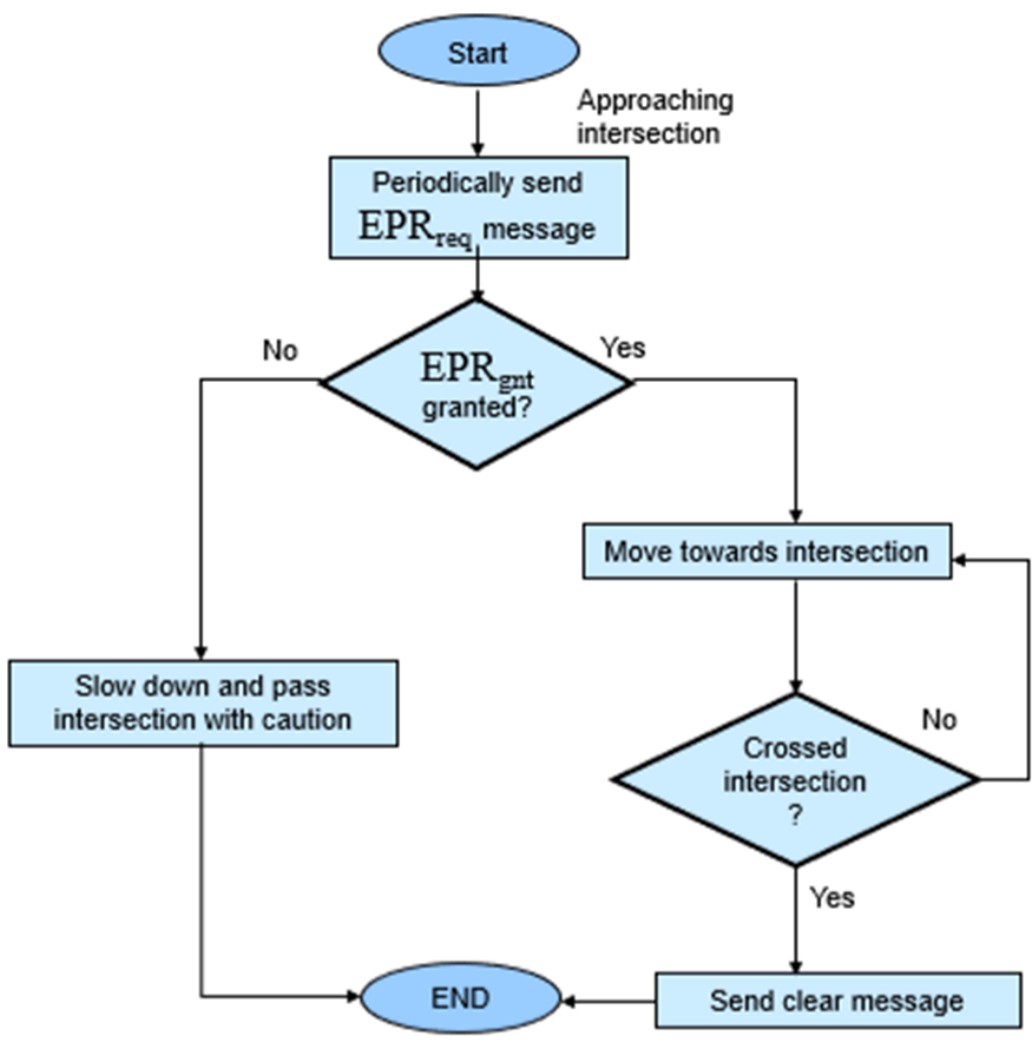

**Figure 4** Emergency vehicle priority flow chart.

### Emergency vehicle priority

The flow chart for emergency vehicle priority is shown in Fig. 4. An EV, upon approaching an intersection, will periodically transmit a beacon message with Emergency Priority Request (ERP) field set to 1. When an AP receives a beacon message with the EPR field set to 1 indicating that this beacon is sent by EV, the AP transmits VTL message (containing green signal) to the lane carrying EV vehicle and transmits red signal to the remaining lanes as shown in Fig. 5. After receiving EPR grant message from AP, the EV moves towards the intersection. After crossing the intersection, the EV transmits clear message to the AP. However, if EPR grant message is not received by the EV, it slows down and pass the intersection with caution.

Algorithm 1 shows the algorithm for EV priority. As an EV approaches an intersection, it periodically broadcasts EPR message (Step2) to the AP installed at the intersection and

waits for the reception of EPRgnt message. Once the EV receives the EPRgnt message, it moves to cross the intersection at a constant speed. After crossing the intersection, the EV transmits CLEAR (CLmsg) message to the AP for resuming normal traffic operations. However, in the event that the AP has not broadcast EPRgnt message or the EV has not received it, the EV shall pass through the intersection with high vigilance.

---

**Algorithm 1** EV Priority Assignment

---

**Input:** $EPR_{gnt}$: EPR granted message
**Output:** $EPR_{rqt}$: EPR request message
$\quad\quad CL_{msg}$ : Clear Message

1: **Step 1: Defining and initialising variables**
2: $\quad EPR_{gnt} =$ Emergency Priority Request Granted
3: $\quad EPR_{rqt} =$ Emergency Priority Request
4: $\quad CL_{msg} =$ Clear Message
5: **Step 2:** Detecting Events
6: **while** 1 **do**
7: $\quad$ regularly transmit $EPR_{rqt}$ message
8: $\quad$ **if** ($EPR_{gnt} = Yes$) **then**
9: $\quad\quad$ -cross the junction
10: $\quad$ **end if**
11: $\quad$ **if** ($EPR_{gnt} = No$) **then**
12: $\quad\quad$ -EV shall pass through intersection with high $\quad$ vigilance
13: $\quad$ **end if**
14: $\quad$ **if** (Crossed intersection = Yes) **then**
15: $\quad\quad$ -transmit $CL_{msg}$
16: $\quad$ **end if**
17: **end while**

---

### *Non-emergency vehicle priority scheme*

In case of non-EVs, the AP computes the traffic density on each approach (explained later in Algorithm 2) and then computes the time for each approach by utilizing Eq. (1):

$$T = \frac{S}{V} \times \frac{N_d}{N_M} \tag{1}$$

where

• S = Distance • V = Velocity • N(d) = Number of Vehicle per direction • N(M) = maximum number of vehicle per direction.

After calculating the time for each approach, the AP assigns green signal to the approach with high vehicular density and red light to remaining approaches as shown in Fig. 6.

In Fig. 6, there are five, three, three, and four vehicles on N-S, W-E, S-N and E-W approach respectively. Therefore, the AP prioritizes the N-S approach due to the high

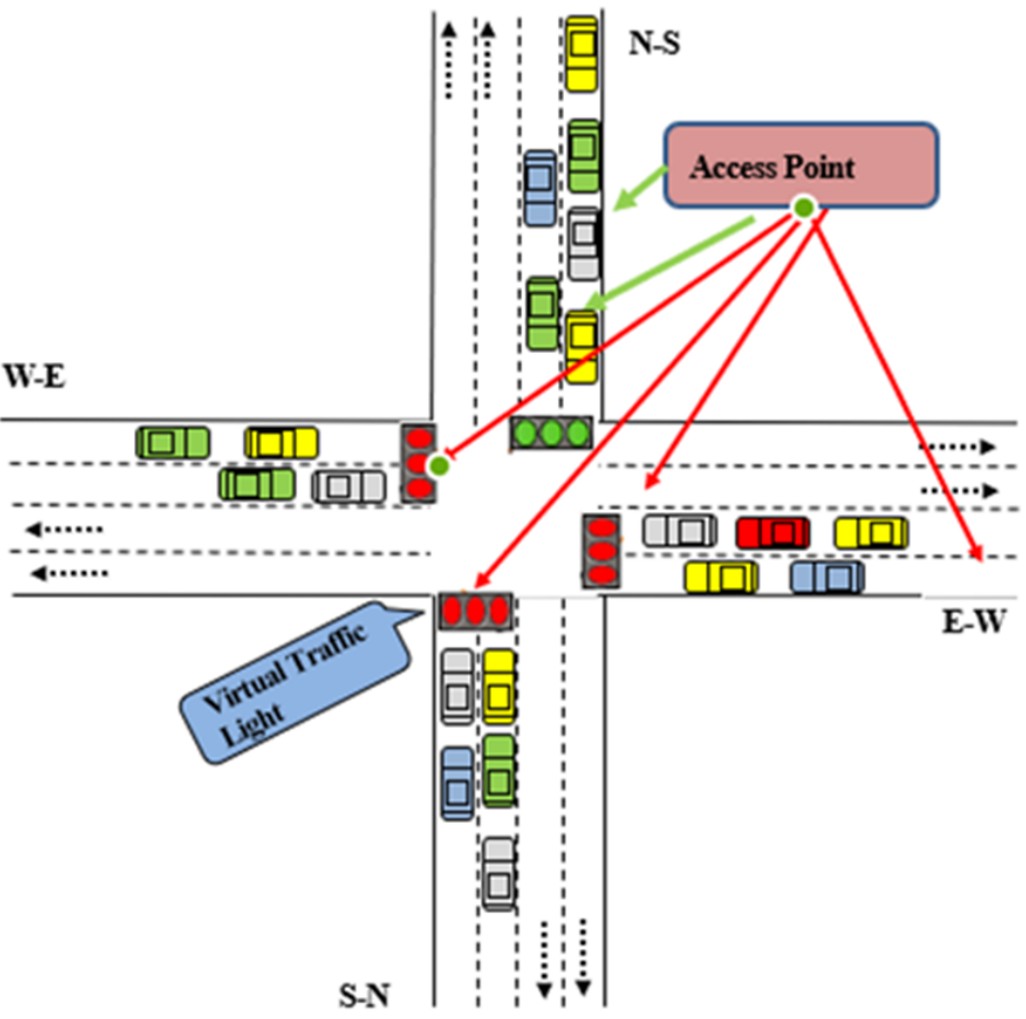

**Figure 6** AP prioritizes the N-S approach due to higher vehicle density.

vehicle density on it and assigns it a green signal for time T computed in Eq. (1). After the N-S approach, the next approach with second highest density is the E-W approach. Therefore, after prioritising N-S approach, the AP assigns green traffic light to E-W approach and red light to all other approaches. The same process continues until all the four approaches are accommodated by the AP.

### Access point

An AP is installed at an intersection to receive beacon messages from vehicles and to transmit VTL information to the vehicle. It is also responsible to calculate vehicle density at each approach and to prioritize EV vehicles. The AP manages the priority for each lane based on number of vehicles available and EV present in each lane. As shown in Fig. 7,

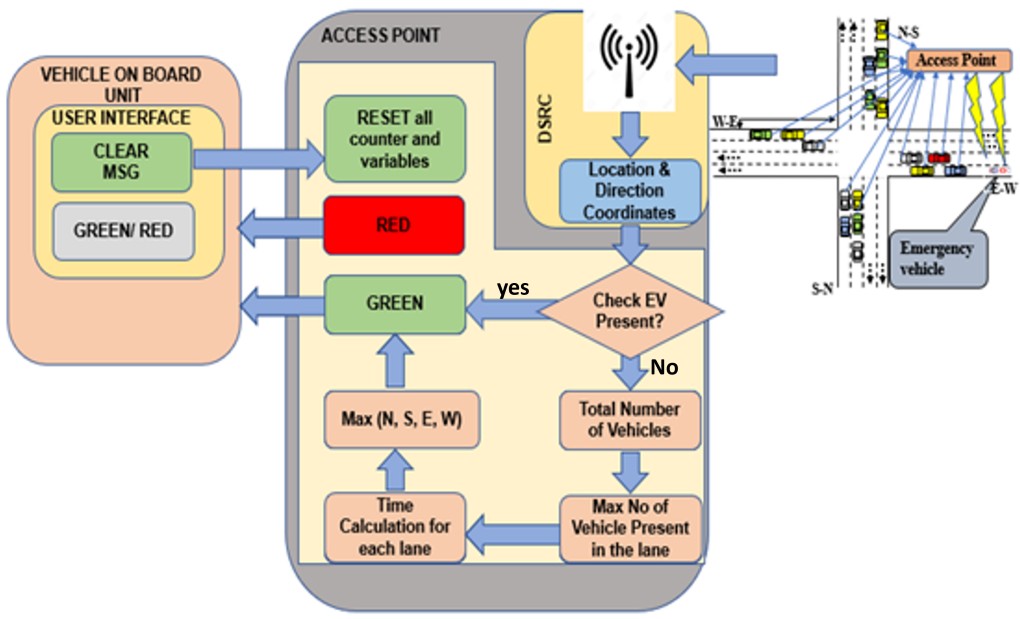

**Figure 7  AP system prioritizes emergency vehicles and high-density lanes.**

the AP receives Location coordinates and direction information of each vehicle present at intersection.

Then, the AP checks for the presence of EV. If present, the AP transmit a green signal to the lane containing EV and red signal to remaining lanes at intersection. However, if EV is not present, the AP finds the total number vehicles present on each lane and also calculates the green signal time for each lane by utilizing Eq. (1). In the first stage, the AP assigns a green signal to a lane containing maximum number of vehicles. Afterwards, the green signal is assigned to the second, third and fourth highest vehicle density lanes respectively.

Algorithm 2 presents the algorithm for AP. The AP receives a beacon message containing vehicles location coordinates, $EPR_{req}$ flag and Clear flag. The AP utilizes vehicles location information to count the number of vehicles in the north ($N_v$), south ($S_v$), east ($E_v$), and west ($W_v$) lanes. If an EPR ($EPR_{req}$) beacon arrives from an EV at intersection, the AP assigns red light to all approaches and assign green light to the respective lane and transmits EPR granted ($EPR_{gnt}$) beacon message to the EV. Once the EV passes through the intersection, it sends a CLmsg to the AP. After reception of $CL_{msg}$, the AP starts regular process. On the other hand, if EV is not detected, then the AP resumes normal operation as follows: The AP finds the maximum vehicle count approach among $N_v$, $S_v$, $E_v$, and $W_v$ and stores the vehicle count in $Max_{veh}$. The AP then calculates green signal time $T_{GS}$ for $Max_{veh}$ lane by utilizing Eq. (1) where $T_{GS}$ represents the time required for the vehicles on a particular lane to cross the intersection. After storing value of timer $T_{GS}$, a green light $GS_{ra}$ signal is sent to the relevant approach. The green light of the traffic signal will remain for $T_{GS}$ duration. Once the $T_{GS}$ expires ($T_{Exp}$), a red light signal $RS_{ra}$ is shown to that side and the vehicle count is set to zero (0) for that approach. This process is continued within

---

**Algorithm 2** Access Point Algorithm

---

**Input:**  $MSG$: Message containing GPS coordinates

$EPR_{rqt}$: EPR request message

$CL_{msg}$: Clear Message

**Output:** $EPR_{gnt}$: EPR granted message

$GS_{ra}$: Green traffic light

$RS_{ra}$: Red light signal

---

1: **Step 1: Defining the variables**

2:      $GS_{ra}$= Green traffic light

3:      $RS_{ra}$= Red light signal

4:      $CL_{msg}$= Clear Message

5:      $Max_{veh}$= Lane containing Maximum vehicles

6:      $N_v, S_v, E_v, W_v$= vehicles count in North, South, East and West approach

7:      $T_{GS}$: Green light time using eq(1).

8:      $T_{Exp}$= $T_{GS}$ Expired

9: **Step 2: Detecting events**

10: **while** 1 **do**

11:      **if** $(EPR_{req} = true)$ **then**

12:          -Transmit $EPR_{gnt}$

13:          -Show $GS_{ra}$

14:      **end if**

15:      **if** $(CL_{msg} = true)$ **then**

16:          -Resume regular process

17:      **end if**

18:      **while** $(N_v, S_v, E_v, W_v)! = 0$ **do**

19:          $Max_{veh}$= MAX$(N_v, S_v, E_v, W_v)$

20:          Calculate and start $T_{GS}$

21:          Show $GS_{ra}$ to $Max_{veh}$ Lane

22:          **if** $T_{Exp} = true$ **then**

23:              Show $RS_{ra}$ to $Max_{veh}$ Lane

24:              Assign $count = 0$ for $Max_{veh}$ Lane

25:          **end if**

26:      **end while**

27: **end while**

---

a loop until all four sides or lanes are addressed and the vehicle counts for all four lanes are set to zero (0). Once the vehicle count of all four lanes is zero, the entire process is repeated.

Figure 8 shows the flow chart for access point. The access point initially receives beacons messages from all the four approaches. After that, it calculates vehicles on each east, west, south and north approach. The access point then checks for the arrival of $EPR_{req}$ message. If the $EPR_{req}$ message is received, the AP transmits EPR granted ($EPR_{gnt}$) beacon message to the EV. The AP will then offer the lane that contains the EV a green light and the other approaches at the junction a red signal. The EV sends a $CL_{msg}$ to the AP after it has passed past the junction. After reception of $CL_{msg}$, the AP resumes normal operation. However, if $EPR_{req}$ message is not received, then the AP resumes normal operation as follows: The AP finds the maximum vehicle count approach among $N_v$, $S_v$, $E_v$, and $W_v$ and stores the vehicle count in $M$. It then calculates green signal time for lane $M$ using Eq. (1) already discussed. Then the green signal is shown to lane $M$ for calculated time period. Once the timer expires, a red signal is shown to lane $M$ and its vehicle count is set to zero (0). Afterwards, the AP finds the second maximum vehicle count approach among remaining three approaches and assigns it a green signal. This process is continued until all four lanes are addressed and the vehicle counts for all four lanes are set to zero (0). Once the vehicle count of all four lanes is zero, the entire process is repeated.

## SIMULATION-BASED PERFORMANCE EVALUATION

In this section, the performance of V2I-VTL is evaluated and compared with a VTL based scheme (*Tonguz & Viriyasitavat, 2016*) by running simulations of different traffic conditions.

### Simulation environment

The proposed V2I-VTL protocol was simulated by using the PTV Vissim (version 9.00) simulator (*Zeidler et al., 2019*). An intersection scenario is considered for both V2I-VTL and VTL schemes. A vehicle can enter intersection from four different directions *i.e.,* west (W), east (E), north (N) and south (S). Each direction has two lanes. During simulations, four sides were taken into account to uphold a changeable vehicle load. In the west direction, 300, 1,000, 1,300, and 1,600 vehicles were deployed during four simulations runs respectively. The maximum vehicles considered at all four approaches during a single cycle are 12 on the average. Similarly, in the south direction, 400, 700, 1,400, and 1,700 vehicles were deployed respectively. In east direction, 500, 800, 1,100, and 1,800. Finally, in north direction, 600, 900, 1,200, and 1,500 vehicles were deployed. The vehicle densities were distributed on all approaches in such a way that roads in each direction observe maximum, normal and minimum densities during a variety of simulation cycles. In the proposed V2I-VTL protocol, the AP was installed at the centre of intersection. It is assumed that in both protocols, each vehicle is equipped with a GPS and a DSRC module with communication range of 200 meter. There are approximately 12 vehicles within the transmission range of AP. The vehicles cross the intersection at speed of 6 km/hr. Table 1 summarizes the VISSIM's simulation parameters.

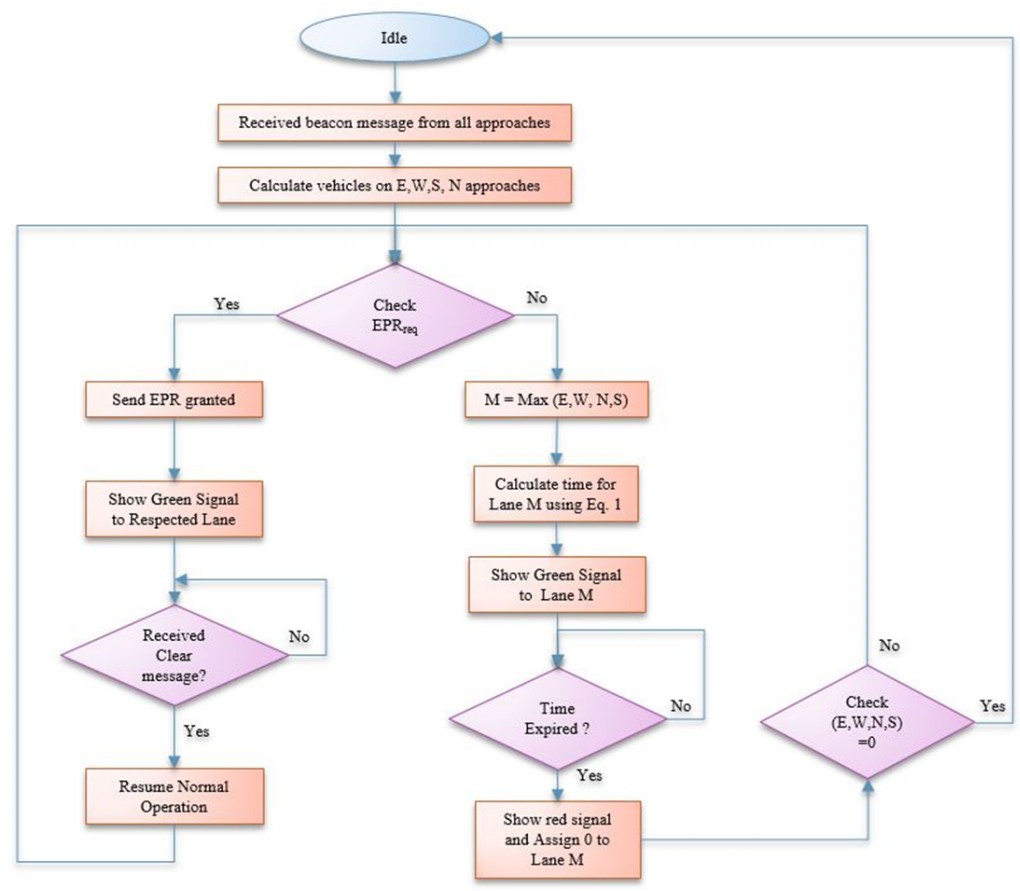

**Figure 8  Flow chart for access point.**

**Table 1  Simulation parameters.**

| Parameters | Value |
| --- | --- |
| Simulation scenario | Intersection/Junction |
| Number of lanes | 2 |
| Vehicle density (W) | 300, 1,000, 1,300, 1,600 |
| Vehicle density (S) | 400, 700, 1,400, 1,700 |
| Vehicle density (E) | 500, 800, 1,100, 1,800 |
| Vehicle density (N) | 600, 900, 1,200, 1,500 |
| Vehicle velocity | 6 km/h |
| VTL DSRC range | 200 meter |
| Simulation duration | 10,000 s |

Moreover, in this study, we investigated $CO_2$ emissions by co-simulating VISSIM and the IPG CarMaker. To realistically model individual vehicle dynamics, the vehicle behavior (such as velocity profiles, acceleration, deceleration, and braking forces) for different vehicle types (light-duty, heavy-duty, and electric vehicles) were modeled in IPG CarMaker.

**Table 2  Summary of IPG CarMaker simulation parameters.**

| Parameters | Value |
| --- | --- |
| Light-duty vehicle | 1,500 kg |
| Heavy-duty vehicle | 7,500 kg |
| Speed at intersection | Modeled dynamically |
| Powertrain types | Internal combustion engine (light & heavy vehicles) |
| Emission estimation model | Vehicle-specific power (VSP) and fuel consumption |
| Acceleration/Deceleration | Detailed dynamics, including engine load, fuel consumption, and rolling resistance |
| CO2 emission factors (Light-duty) | Based on fuel consumption |
| CO2 emission factors (Heavy-duty) | Diesel engine |

CarMaker provides a detailed, physics-based simulation environment for Powertrain behavior including fuel consumption and emissions specific to internal combustion engines (ICE) and electric powertrains. Moreover, vehicle-specific dynamics including different vehicle masses, aerodynamics, and rolling resistance for light-duty, heavy-duty, and electric vehicles were considered to improve emission accuracy. The following parameters were modeled using CarMaker for more realistic emission calculations:

- **Engine load:** CarMaker accounts for the engine load under various conditions, including idling, acceleration, and deceleration, based on the actual power demand from the vehicle.
- **Fuel consumption** Fuel consumption data was computed dynamically based on the vehicle's speed, engine load, and resistance factors.
- **CO2 emission factors:** CO2 emissions were calcuated using the fuel consumption rate and the carbon content of the fuel, providing vehicle-specific emission outputs (in grams of CO2 per kilometer).

For light-duty vehicles, typical parameters such as vehicle weight (1,500 kg), engine type, and aerodynamic properties were used. For heavy-duty vehicles (7,500 kg), a diesel powertrain was modeled with appropriate rolling and air resistance coefficients. Electric vehicles were assumed to have zero tailpipe emissions, but their energy consumption was modeled to estimate the indirect emissions depending on the electricity grid mix. A co-simulation framework was set up between VISSIM and CarMaker, where VISSIM provided traffic scenarios and vehicle trajectories (speed, acceleration, braking events), while CarMaker computed the corresponding CO2 emissions based on vehicle-specific dynamics. Table 2 summarizes the Carmaker's simulation parameters.

## Performance metrics

The following metrics were used to compare the performance of V2I-VTL with VTL scheme.

- **Lane opening time (LOT):** Time at which a particular approach is shown green traffic signal to let vehicles cross the intersection.

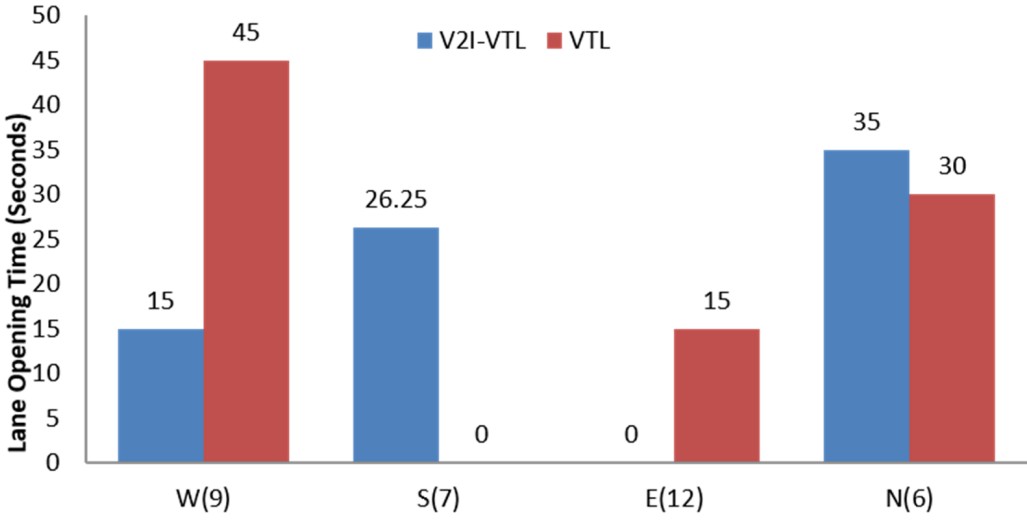

**Figure 9** Lane direction and vehicle count in the first cycle.

- **Total delay experienced (TD):** The total sum of all cycle delays that occurred throughout the simulation, where a cycle delay is the amount of time needed for one cycle to be completed for each of the four approaches at the intersection.
- **$CO_2$ emissions per cycle (CEC):** The $CO_2$ emissions per cycle represent the average amount of $CO_2$ emitted during one complete cycle of the traffic signal.

### Lane opening time (LOT)

Figure 9 compares V2I-VTL and VTL in terms of LOT based on vehicle density in each direction. LOT during 1st cycle are shown in Fig. 9. In VTL, lanes are opened after a fixed duration of 15 s. For that reason, the leader vehicle shows green signal to the south direction on zeroth position and red signal to the rest. Then, after 15 s, the east direction is shown the green signal. Similarly, the north and west directions are opened after 30 and 45 s, respectively. However, LOT in V2I-VTL depends on the density of vehicles in each direction. It is shown in Fig. 9 that there are 12 vehicles in east direction *i.e.,* E(12) hence, it is opened first and shown green signal by the AP because this approach has more vehicles (12) as compared with other approaches. In the proposed V2I-VTL, the duration of the green signal, as explained earlier in Eq. (1), is determined based on the number of vehicles in each direction. Afterwards, the west, south, and north directions are opened having nine, seven and six vehicles respectively. In V2I-VTL, owing to the variable lane-opening times by the AP, the LOT for the last side, *i.e.,* north, is 35 s, in comparison to the 45 s LOT in case of VTL.

Figure 10 compares LOT for VTL and V2I-VTL based on vehicle density in each direction in the second cycle. In case of VTL, the east approach E(4) is shown green signal at zeroth second by the leader vehicle. Then, after 15 s, the north lane is opened. Similarly, the west and south approaches are shown green signal after 30 and 45 s, respectively. However,

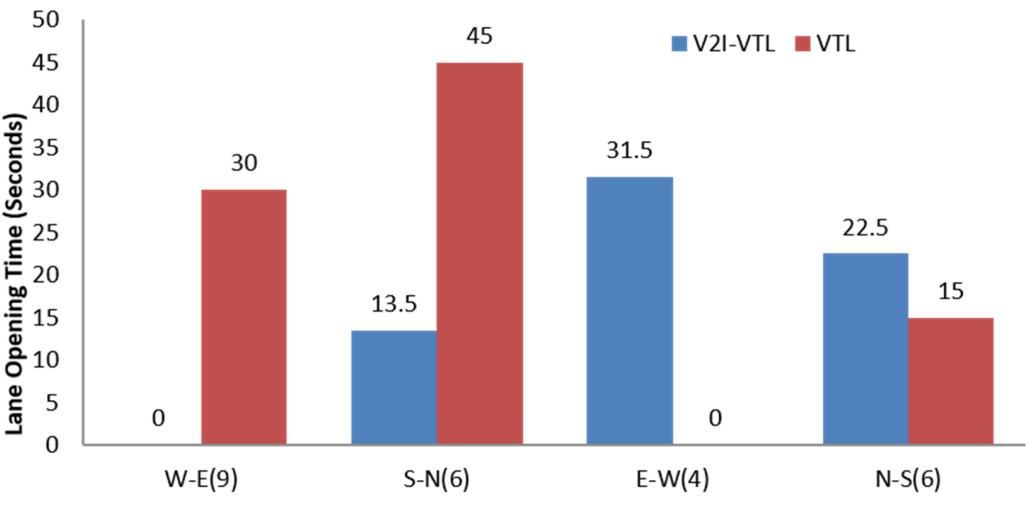

**Figure 10  Lane direction and vehicle count in the second cycle.**

for V2I-VTL, it is shown that west direction is opened first and shown green signal by access point because it has more vehicles W(9) compared with other approaches. Then, the south, north and east directions are opened having six, six, and four vehicles respectively. In V2I-VTL, the lane opening time for the last east approach is 31.5 s, in comparison to 45 s LOT in case of VTL.

Figure 11 evaluates the time for lane opening intended for V2I-VTL and VTL in the third cycle. In case of VTL, the north approach is opened at zeroth second by the leader vehicle. Then, after 15 s, the west direction is opened. Similarly, the south and east approaches are opened after 30 and 45 s, respectively. However, in proposed V2I-VTL, the south direction is opened first and shown green signal by access point because it has more vehicles S(11) compared with other approaches. Then, the east, north and west directions are opened having 10, nine and eight vehicles respectively. In V2I-VTL, the LOT for the last west approach is 37.5 s, compared with the 45 s LOT in case of VTL.

Figure 12 compares the LOT for V2I-VTL and VTL in the fourth cycle. In case of VTL, the west direction is opened at zeroth second by the leader vehicle. Then, after 15 s, the south direction is opened. Similarly, the east and north sides are opened after 30 and 45 s, respectively. However, in V2I-VTL, it is shown that north direction is opened first and shown green signal by AP because it has more vehicles N(12) compared with other approaches. Then, the east, south and west directions are opened having 11, nine and eight vehicles respectively. In V2I-VTL, for the last west approach is 40 s, compared with the 45 s LOT in case of VTL.

In Figs. 9–12, the LOT in VTL for the last approach is 45 s during all four cycles. This is because the green signal time for each approach is 15 s in VTL and does not depend on the vehicle density. However, in V2I-VTL, LOT of last approach changes during all four cycles

and depends upon vehicle density on particular approach. In V2I-VTL, the maximum lane opening time was 40 s and the minimum was 31.5 s. Therefore, the lanes having high vehicular density are assigned high priority in V2I-VTL system, which decreases vehicle waiting time at intersection and avoids the formation of extensive queues.

### Comparison using T-Test

The cycle times for 10 different cycles were selected for both VTL and V2I-VTL protocols. For VTL, the cycle time is fixed, with each cycle taking exactly 60 s.

- **VTL cycle times**: [60, 60, 60, 60, 60, 60, 60, 60, 60, 60]
- **V2I-VTL cycle times**: [47.5, 47.5, 45.0, 31.25, 52.5, 37.5, 37.5, 38.75, 42.5, 35.0]

- **Null hypothesis ($H_0$)**: The mean cycle times for the VTL and V2I-VTL protocols are the same.
- **Alternative hypothesis ($H_1$)**: The mean cycle times for the VTL and V2I-VTL protocols are different.

For the VTL protocol, the mean cycle time ($\bar{X}_1$) is 60 s with a standard deviation ($s_1$) of 0 s, as all cycle times are identical. For the V2I-VTL protocol, the mean cycle time ($\bar{X}_2$) is 41.5 s with a standard deviation ($s_2$) of 6.61 s, reflecting the variability in cycle times based on the dynamic vehicle load at the intersection.

We used an independent $t$-test to compare the two protocols. The formula for the t-statistic is:

$$t = \frac{\bar{X}_1 - \bar{X}_2}{\sqrt{\frac{s_1^2}{n_1} + \frac{s_2^2}{n_2}}}$$

where:

- $\bar{X}_1$ and $\bar{X}_2$ are the means of the two groups.
- $s_1^2$ and $s_2^2$ are the variances of the two groups.
- $n_1 = n_2 = 10$ (number of cycles).

The t-statistic is 8.85, and the $p$-value is $5.64 \times 10^{-8}$. The $p$-value is extremely small ($5.64 \times 10^{-8}$), which is far below the standard significance level of 0.05. This means that we can reject the null hypothesis and conclude that there is a statistically significant difference between the cycle times of the VTL and V2I-VTL protocols. Additionally, the t-statistic of 8.85 indicates a strong difference between the two groups, with V2I-VTL showing significantly shorter cycle times compared to the VTL protocol.

### Lane opening time for single emergency vehicle (EV)

Figure 13 shows the Lane opening times when an EV arrives. In case of VTL, the east side is opened in the zeroth second as the leader vehicle gives green signal to the east. Then, north and west sides are opened on 15 and 30 s, respectively. After the west direction, the last direction to give green light is the south direction. Meanwhile, an EV, *e.g.*, an ambulance arrives on the north approach when the AP is busy with west approach. In the VTL scheme, the north approach will be given priority again after completing the west approach at 45 s.

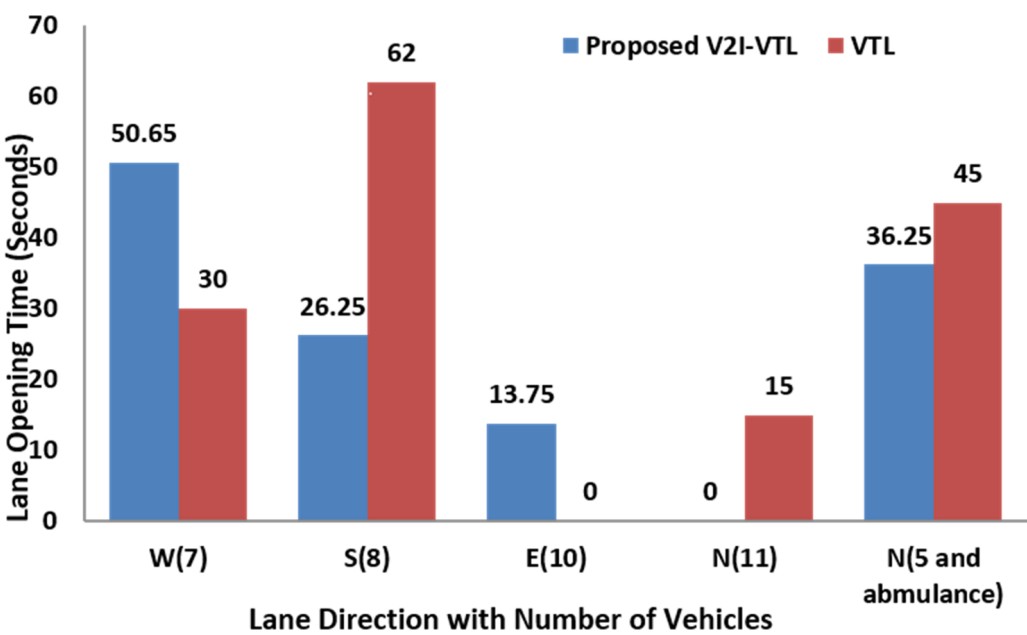

**Figure 13  Lane opening time while EV arrives.**

The north approach will be opened again at 45th second instead of opening the south approach, thus reducing the waiting time of EV at intersection. The north direction is opened again by VTL because an ambulance arrived, therefore it must be given priority by opening the north lane at 45 s exactly 15 s later than lane opening time of west approach at 30 s. The north lane will remain open until the ambulance crosses the intersection and sends a 'Clear' message. The south direction is opened after the EV exits the intersection at 62 s.

However, in V2I-VTL, the LOT for any approach totally depends on the number of vehicle available on a particular direction. In Fig. 13, it is shown that the north side is opened at zeroth second as there are maximum 11 vehicles present in north direction. Then, the east and south directions are opened after 13.75 and 26.25 s with 10 and eight vehicles, respectively. Following the south side, the next side to open is west side with seven vehicles. Meanwhile, an ambulance arrives in the north direction with five additional vehicles. As discussed earlier in the flow chart of AP in Fig. 8, the AP, after showing red signal to the particular lane $M$, checks for the arrival of EPR message from EV. If EPR is received by the AP, then it interrupts the normal operation and prioritizes the EV. Therefore, in Fig. 13, the AP delays the opening of west approach and prioritizes the EV approaching at north direction. The AP opens the north approach at 36.25 s instead of the west approach, thereby reducing the waiting time of EV at the intersection. Therefore, in Fig. 13, lane opening time for EV in case of V2I-VTL is 36.25 s whereas in case of VTL it is 45 s. This is because in VTL, a particular direction remains open for traffic flow for a fixed duration of 15 s, thereby increasing the waiting times of remaining approaches.

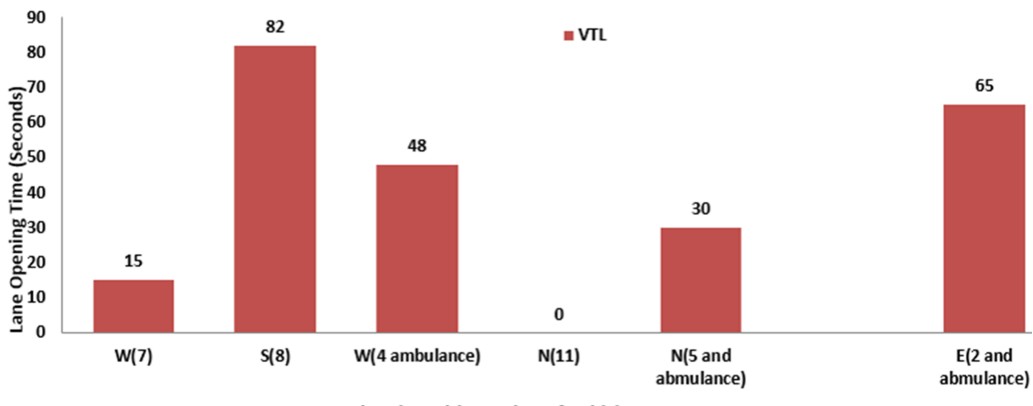

**Figure 14** Lane opening time when multiple EVs arrives in VTL.

### Lane opening time for multiple emergency vehicles

Figure 14 shows the lane opening times when multiple ambulances arrive at an intersection under the VTL scheme. Initially, the north side is opened at the 0th second as there are 11 vehicles waiting, including an ambulance. The west direction is opened next at 15 s with seven vehicles. However, while the west approach is in progress, another ambulance arrives on the north approach. As the VTL scheme prioritizes emergency vehicles, the north direction is opened again at 30 s to allow the ambulance to cross the intersection, ensuring minimal waiting time. After the north side is cleared, another ambulance arrives on the west side, prompting the VTL system to reopen the west approach at 48 s, overriding the initial sequence. Once the ambulance has cleared the west approach, an additional ambulance arrives on the east direction. The east lane is then opened at 65 s to prioritize this emergency vehicle.

Finally, after all the emergency vehicles have been accommodated, the south side is opened at 82 s, marking the end of the cycle. This adjusted sequence illustrates how the VTL scheme changes its lane opening order to prioritize emergency vehicles, delaying the south approach until all ambulances have passed through the intersection. In the original sequence, south would have opened earlier, but the ambulances on the north, west, and east sides caused delays, with south being opened last.

However, in the proposed V2I-VTL scheme, lane openings are determined based on the number of vehicles in each direction, with priority given to EVs when they arrive as shown in Fig. 15. The north side opens first at 0 s, as it has the highest vehicle density with 11 vehicles. The east side follows, opening at 13.75 s, handling 10 vehicles. When an ambulance arrives at the north approach, the system prioritizes this and reopens the north lane at 26.25 s, minimizing the waiting time for the emergency vehicle. After the ambulance on the north side has cleared, the west side, which has seven vehicles and an ambulance, opens at 40.65 s. This ensures that the ambulance on the west side is given priority. When

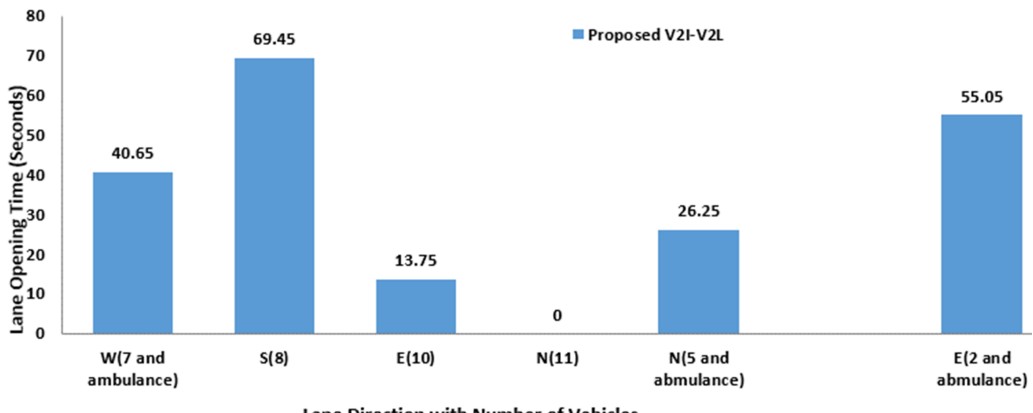

**Figure 15** Lane opening time when multiple EVs arrives in V2I-VTL.

another ambulance arrives at the east side with two vehicles, the system reopens the east lane at 55.05 s to accommodate it.

Finally, after all ambulances have passed through the intersection, the south side opens last at 69.45 s. This approach contrasts with the fixed lane opening times in the VTL scheme, where the south approach opens last at 82 s. In the proposed V2I-VTL system, the south approach opens earlier at 69.45 s, as the system dynamically prioritizes emergency vehicles and adjusts the lane openings in real-time. This results in a more efficient management of traffic, reducing delays caused by emergency vehicles and improving overall flow at the intersection.

### Total delay experienced (TDE)

Figure 16 compares the total delay experienced by both the protocols with respect to vehicles density. The figure shows the delays experienced by vehicles in 10 cycles including EVs and non-EVs. It is clear that the delay increases linearly with the density for both schemes. In VTL scheme, the cycle delay in which all four directions are served is always 60 s. However, in the V2I-VTL methodology, the cycle delay is random and depends on the vehicle density in each approach. Therefore, the total delay experienced by vehicles in proposed scheme is smaller when compared with VTL scheme. The proposed V2I-VTL scheme calculates the traffic light signal time based on the density of vehicles in each direction. Therefore, it is evident that that the V2I-VTL protocol can trim down the delays of EV under light and heavy traffic conditions and these delays are always smaller than the delays experienced by EV when VTL scheme is used.

### CO$_2$ emissions per cycle (CEC)

Figure 17 presents an analysis of CO$_2$ emissions for both V2I-VTL and VTL schemes, comparing light-duty, heavy-duty, and electric vehicles across a complete intersection cycle, which includes the phases of idling, acceleration, deceleration, and steady driving for each vehicle type (light-duty, heavy-duty, and electric vehicles). The results reflect the differences between the VTL and V2I-VTL protocols. For light-duty vehicles, the total

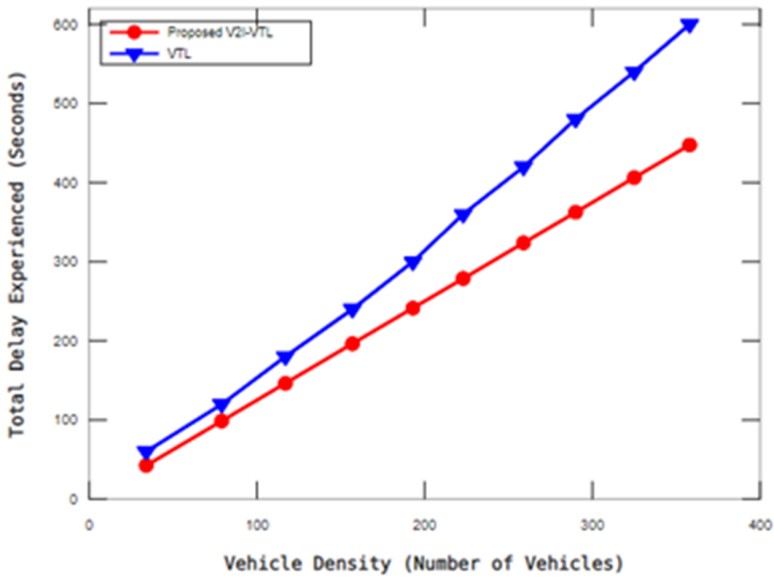

**Figure 16** Total delay experienced.

$CO_2$ emissions for the VTL protocol are approximately 70 grams per cycle, whereas under the V2I-VTL protocol, emissions drop significantly to about 20 grams. This represents a reduction of around 64%, attributed to the dynamic adaptation of signal timing, which reduces unnecessary idling and the number of stops and starts. For heavy-duty vehicles, $CO_2$ emissions for the VTL protocol are around 350 grams per cycle, which is substantially higher due to the larger engine load and fuel consumption. Under the V2I-VTL protocol, emissions reduce to approximately 150 grams, representing a 57% reduction compared to the VTL protocol. This significant reduction is due to the reduced idle time and more efficient acceleration/deceleration cycles in the V2I-VTL system.

Electric vehicles exhibit zero direct tailpipe $CO_2$ emissions under both protocols, as expected, though this does not account for indirect emissions related to the energy source used for charging. Overall, the V2I-VTL protocol consistently reduces $CO_2$ emissions across both light-duty and heavy-duty vehicles compared to the traditional VTL protocol. The reduction in emissions ranges from 57% to 64%, highlighting the effectiveness of V2I-VTL in optimizing traffic flow at intersections. Heavy-duty vehicles, due to their larger engine size and fuel consumption, contribute more significantly to $CO_2$ emissions, underscoring the importance of focusing on emission reduction strategies for these vehicles. Implementing V2I-VTL can significantly reduce emissions in congested urban areas, especially where heavy traffic or a high volume of heavy-duty vehicles is present.

Building upon the simulated success of the proposed system, we have taken a significant leap forward by integrating and testing the proposed system in a real-world environment. This crucial phase involved the meticulous installation and calibration of the necessary hardware at a busy intersection, thereby transitioning from theoretical models to tangible, operational technology. The hardware implementation serves as a testament to the practical

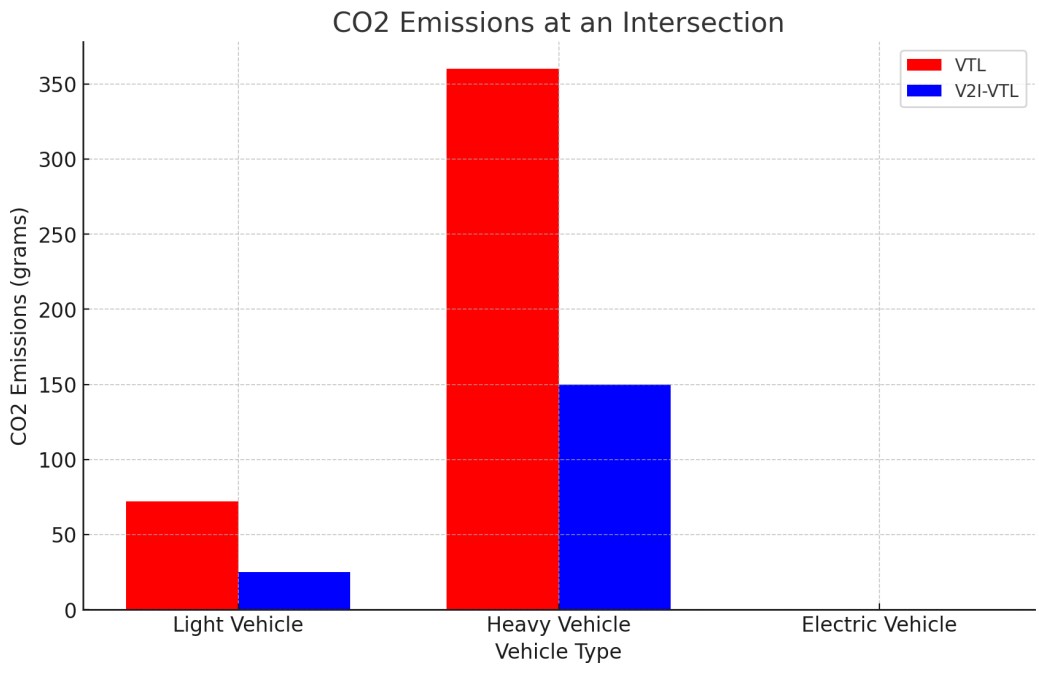

**Figure 17** $CO_2$ emission.

viability of our solution, offering a more comprehensive evaluation of its effectiveness in managing traffic flow and prioritizing emergency vehicles. The following paragraphs will detail the hardware setup, the challenges encountered during installation, and the subsequent impact on traffic efficiency and emergency response times.

## REAL-WORLD HARDWARE VALIDATION

Figure 18 depicts a system architecture or block diagram for the proposed traffic management system. This system appears to integrate various components to manage traffic flow, particularly focusing on emergency and vehicle units at intersections. Here's a detailed breakdown of the components and their functions:

- **GPS:** A GPS module, short for Global Positioning System module, is a device that receives signals from satellites orbiting the Earth to determine the device's geographical location.
- **ESP8266 NodeMCU:** A low-cost Wi-Fi microchip with full TCP/IP stack and microcontroller capability, which recevies the GPS coordinates of the vehicle and transmit it to the intersection controller.
- **GSM module:** Stands for Global System for Mobile Communications. It is used for for sending alerts from emergency vehicle to intersection controller.
- **Emergency unit:** A dedicated system or module designed to detect and respond to emergency situations, by altering intersection controller to clear paths for emergency vehicles.

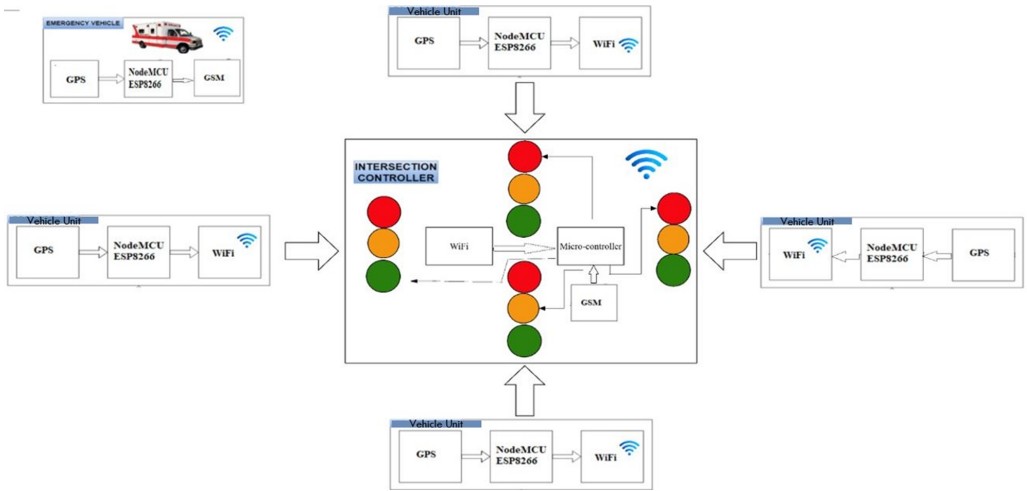

**Figure 18  Block diagram of installed hardware architecture.**

- **Vehicle unit:** This refers to a component within each vehicle that communicates with the intersection controller, to receive traffic light status.
- **Intersection unit:** The main control unit at each intersection, which coordinates the traffic lights and communicates with the emergency and vehicle units to optimize traffic flow.

The depicted system in Fig. 18 is designed to promptly respond to emergency vehicles (EVs) at intersections, ensuring their expedited passage while maintaining orderly traffic flow for non-emergency vehicles. As illustrated in Figure 3.1, the system comprises two principal components: the vehicle on board unit (OBU) and the Intersection Controller (IC), the latter also serving as an access point. The OBU, equipped on all vehicles, is tasked with broadcasting the vehicle's GPS coordinates to the IC. This includes a beacon message from each OBU that conveys the precise location of the vehicle. Emergency vehicles, such as fire engines, ambulances, police vans, and military transport, are similarly outfitted with OBUs to transmit their positional data. Concurrently, the IC, strategically located at the intersection, is responsible for aggregating this locational data and vigilantly monitoring emergency vehicle requests to prioritize their movement through the intersection. The IC further manages the flow of regular traffic by assessing the traffic density in each direction, details of which are elaborated subsequently.

## Circuit diagram

Figure 19 presents the circuit diagram of the proposed system. Vehicles designated for emergencies, such as ambulances, fire trucks, and police vehicles, are outfitted with a specialized Ambulance Unit. This unit is comprised of a GPS module for location tracking, a controller to process data, and a GSM module for communication. The system is designed to give precedence to these emergency vehicles by responding to Emergency Priority Requests (EPR). The Ambulance Unit actively monitors the vehicle's proximity to the intersection, and upon reaching a threshold distance of less than one

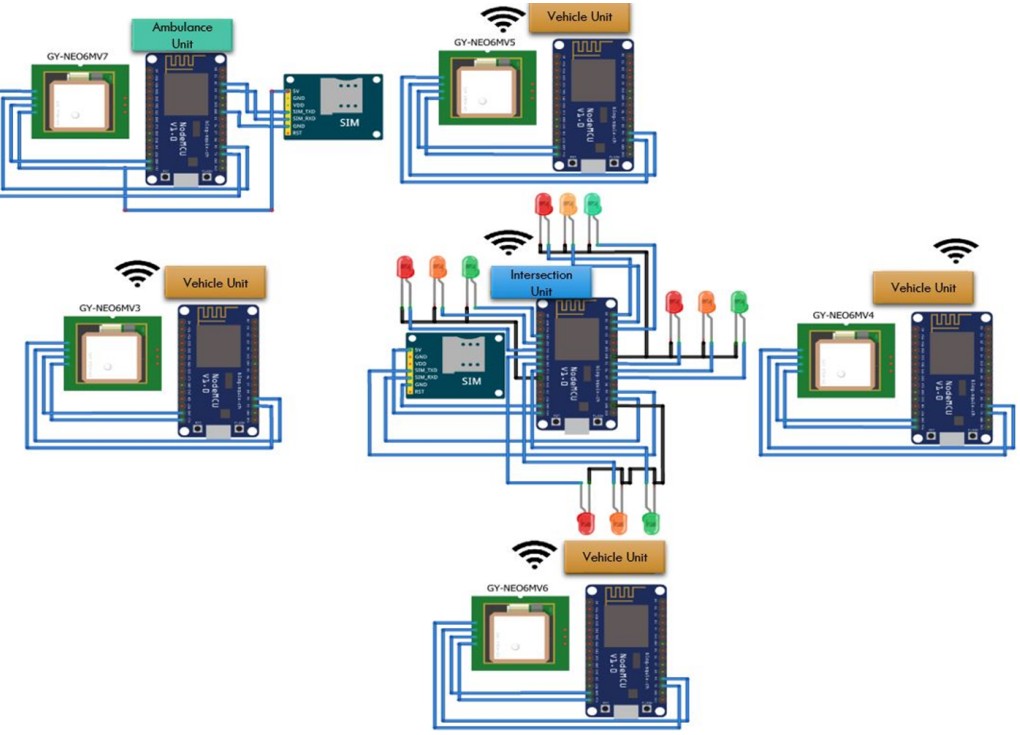

**Figure 19** **Circuit diagram of the proposed system.**

kilometer, it transmits an EPR message to the Intersection Controller (IC). After the emergency vehicle has successfully navigated the intersection, it communicates a 'clear' signal to the IC, prompting the traffic system to resume normal operations. The vehicle unit (VU) initiates searching for available WiFi networks and, upon detecting the WiFi access point of an IC, it connects to this network. Once connected, the VU sends a beacon message to the IC, which includes the vehicle's ID and its GPS coordinates, enabling the IC to identify and locate the vehicle as it enters the range of the WiFi network.

## Intersection controller flowchart

The flowchart in Fig. 20 outlines the algorithm for an IC, which is a key component in managing traffic flow at intersections. The IC algorithm functions as follows:

- **Emergency vehicle priority**: The IC prioritizes emergency vehicles by responding to their priority requests, ensuring they can pass through the intersection swiftly.
- **Beacon message reception**: The IC receives beacon messages from all vehicles approaching the intersection. These messages contain the vehicle's ID and GPS coordinates.
- **Vehicle location and count**: Using the GPS coordinates from the beacon messages, the IC determines the location of each vehicle and increments the vehicle count for each lane accordingly.

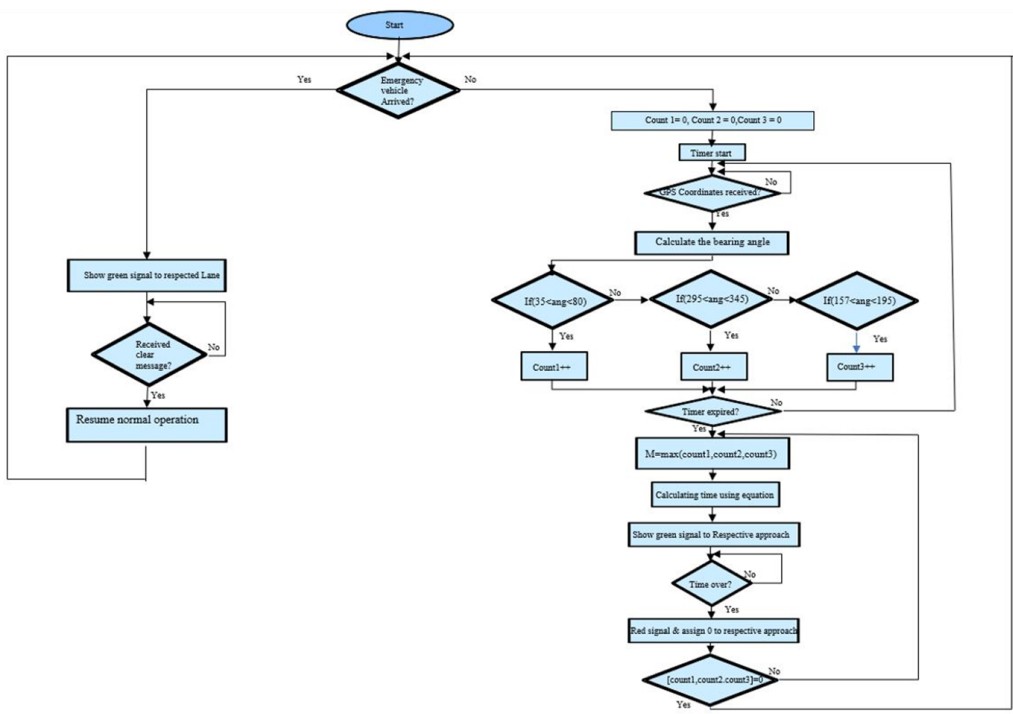

**Figure 20   Flowchart of intersection unit.**

- **Traffic signal management**: The IC evaluates which lane has the maximum number of vehicles and calculates the duration for the green signal based on this count. It then displays the green signal to the lane with the highest vehicle count.
- **Signal transition**: Once the calculated on-time for the green signal expires, the IC switches the signal to red for that lane and repeats the process for the next lane with the highest vehicle count.

This flowchart ensures an efficient and responsive traffic management system, capable of adapting to real-time traffic conditions and prioritizing emergency responses when necessary. The use of beacon messages for vehicle identification and location tracking allows for a dynamic and automated approach to traffic signal control.

## Finding angle and direction of vehicle

The intersection's system captures the coordinates of vehicles coming from all directions. The key challenge lies in determining the vehicle's approach angle or direction. This can be ascertained using the Azimuth angle, also known as the Bearing angle. The Azimuth angle, which measures the angle between the meridian (a line towards true north) and a point of interest, ranges from 0° to 360°, with true north commonly set at 0°. A slight variation exists between true north and magnetic north due to the earth's magnetic elements, which deflect the compass needle towards magnetic north. True north can be pinpointed with electronic instruments. The formula provided in Eq. (2). computes the bearing angle of a vehicle as it nears the intersection, relative to the intersection's true north. Here, $\phi_1$

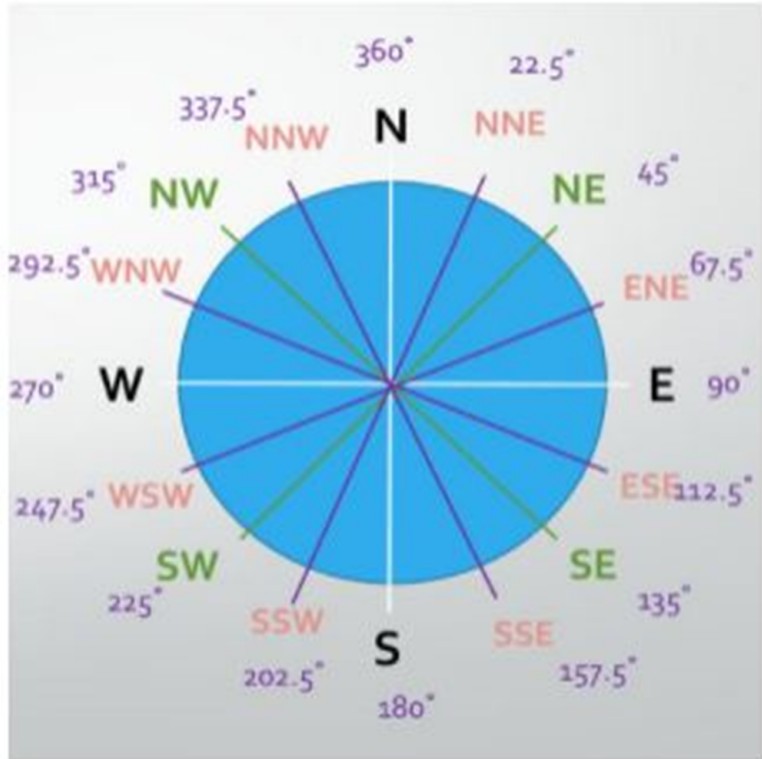

**Figure 21 Cardinal, ordinal and tertiary directions.**

represents the latitude of the intersection point, $\phi_2$ is the latitude of the vehicle, and $\Delta\lambda$ is the longitudinal difference between the vehicle and the intersection point.

$$\theta = \text{atan2}(\sin\Delta\lambda \cdot \cos\phi_2, \quad \cos\phi_1 \cdot \sin\phi_2 - \sin\phi_1 \cdot \cos\phi_2 \cdot \cos\Delta\lambda). \tag{2}$$

By utilizing Eq. (2), the vehicles' azimuth or bearing angle can be calculated. Once the bearing angle is ascertained, pinpointing the direction becomes straightforward. Figure 21 illustrates the cardinal, ordinal, and tertiary directions (*Reading & understanding direction on maps, 2020*). The fundamental cardinal directions comprise north, south, east, and west. In contrast, numerous ordinal and tertiary directions exist. Ordinal directions lie between the cardinal points, whereas tertiary directions are situated between the ordinal points.

### Deployment scenario

The system's deployment at the COMSATS University Islamabad, Attock campus intersection is characterized by three distinct approaches. Table 3 outlines the hardware implementation parameters, revealing that the northern approach accommodates three vehicles, while the western and southern approaches handle two and one vehicle respectively. The vehicles are assumed to travel at a velocity of 7.2 km/h, with a standardized distance assumption of 15 m. These specifications provide a comprehensive overview of the system's hardware configuration within the specified educational setting.

| Table 3 | Implementation parameters. | |
|---|---|---|
| **Parameters** | | **Value** |
| No. of approaches | | 3 |
| No. of vehicles (N) | | 3 |
| No. of vehicles (W) | | 2 |
| No. of vehicles (S) | | 1 |
| Velocity of the vehicles | | 7.2 km/h |
| Distance assumed | | 15 m |

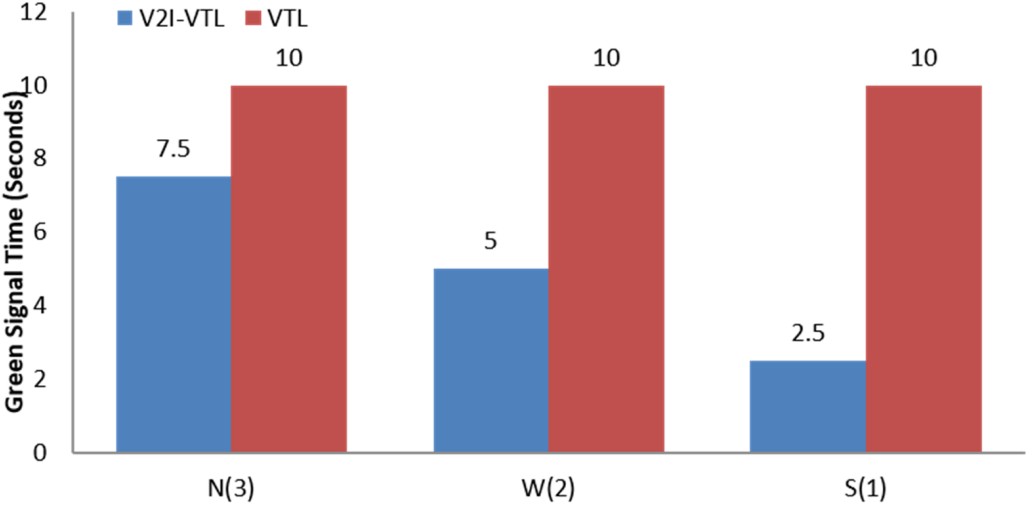

**Figure 22** Green signal time for each approach.

## Green signal time

Figure 22 contrasts the green signal duration allocated by the proposed adaptive traffic control system with that of a conventional fixed time slot system. The latter uniformly distributes a green signal duration of 10 s to each approach, irrespective of vehicular presence, culminating in a total delay of 30 s. Conversely, the proposed system dynamically adjusts the green signal times based on Eq. (1), tailored to the actual traffic conditions, resulting in a significantly reduced total delay of 15 s. This comparison underscores the proposed system's superior efficiency, halving the delay time and optimizing traffic flow.

## Lane opening time

Lane opening time is a crucial metric for traffic management, representing the moment when a specific direction is granted a green signal. This timing is directly proportional to vehicle density; higher vehicle counts lead to increased lane opening times. According to Fig. 23, the proposed system allocates lane opening times of 0, 7.5, and 12.5 s for the north, west and south approaches, respectively. In contrast, the fixed time slot system assigns 0, 10, and 20 s for these directions. The proposed system's reduced lane opening times, when

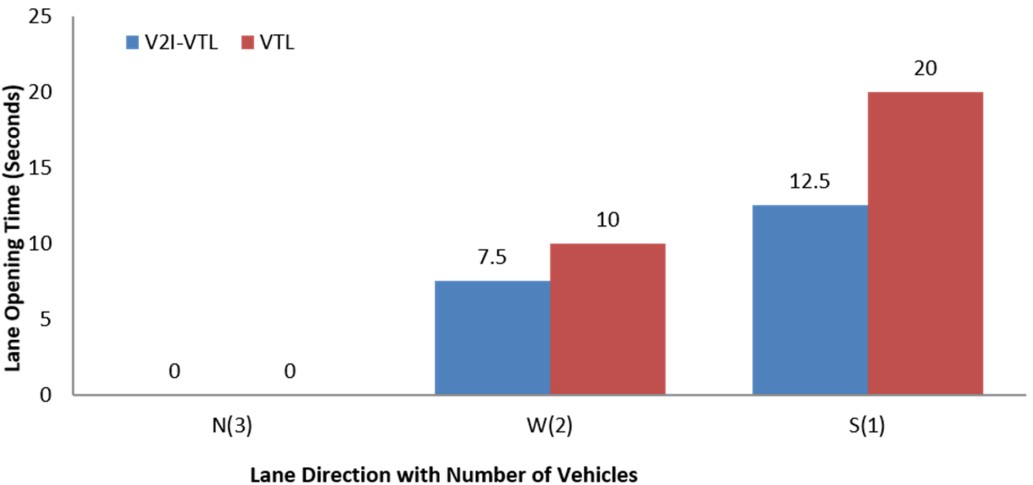

**Figure 23** Lane opening time.

compared to the fixed time slot system, highlight its efficiency. Therefore, the proposed system outperforms the fixed time slot system regarding both green signal duration and lane opening times.

## Lane opening time for emergency vehicles

This section focuses on the lane opening time for emergency vehicles, which is a key factor in traffic flow management for such vehicles.

### Case 1: Emergency vehicle approaching from north

Imagine a situation at an intersection where the northbound approach has three vehicles, the westbound has two, and the southbound has only one. As depicted in Fig. 24, the traffic signal system prioritizes the northbound approach due to its higher vehicle count, followed by the westbound approach. During the westbound signal, an emergency vehicle approaches from the north. The system prioritizes the emergency vehicle, switching the northbound signal to green at 12.5 s immediately after the westbound. Once the emergency vehicle has cleared, the signal for the southbound approach turns green, completing the cycle before restarting with the approach that has the most vehicles.

In contrast, under a fixed time slot system, the emergency vehicle approaching from the north receives no priority, as shown in Fig. 24. The signal transitions from westbound to southbound at 20 s, and only after the southbound cycle does the northbound approach receive a green signal at 30 s. However, in the proposed system, when an emergency vehicle approaches from the north, the northbound signal changes to green at 12.5 s, immediately following the westbound signal.

In the emergency vehicle priority system, it is presumed that such vehicles send out a priority signal when they are 500 m from the traffic intersection. For the purpose of safety, these vehicles are expected to maintain speeds between 50 and 80 km/h. Calculating the green signal duration using the distance formula given in Eq. (3) with a speed of 50 km/h

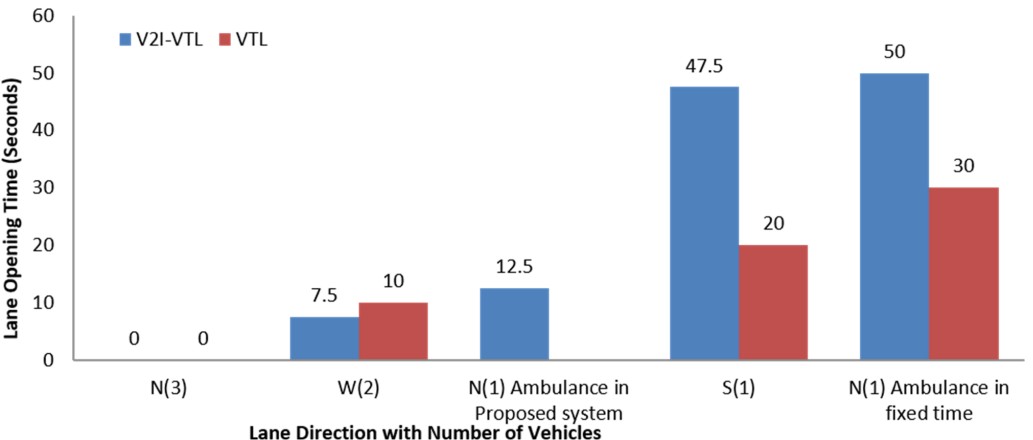

**Figure 24** Lane opening time of emergency vehicle approaching from north.

(equivalent to 14 m/s) and a distance of 500 m, the result is approximately 35.71 s. This figure is then rounded to 35 s. Therefore, the total time for the south lane to remain open is determined to be 47.5 s, which includes an initial 12.5 s plus the calculated green signal time.

$$t = \frac{S}{v}. \tag{3}$$

***Case 2: Emergency vehicle approaching from west***
Envision a different situation at an intersection where there are three vehicles waiting on the north side, two on the west, and one on the south. While the southbound traffic is being allowed to pass, an emergency vehicle from the west sends a priority request. The advanced traffic signal system then gives precedence to the westbound emergency vehicle, granting it a green signal immediately after the southbound traffic. Following this, the system proceeds to the next cycle, signaling green for the northbound traffic. Under the same assumptions as in the first case, the green signal duration for the emergency vehicle in the new system is calculated. The northbound lane's opening time is determined to be 15 s plus 35 s, which equals 50 s. This timing is illustrated in Fig. 25, where the lane opening time for the emergency vehicle approaching from the west is shown. In this updated system, the westbound emergency vehicle's lane opens at 15 s, in contrast to the 40 s it would take under a fixed time slot system.

## CONCLUSION

In this research work, a novel traffic management scheme named V2I-VTL is designed to optimize vehicle transit through intersections while prioritizing emergency vehicles. Utilizing an access point (AP) installed at intersections, the scheme communicates with both emergency and non-emergency vehicles *via* a Dedicated Short-Range Communication (DSRC) module, enabling real-time assessment of traffic density on each approach. Simulation results demonstrate the efficacy of the V2I-VTL scheme in reducing intersection

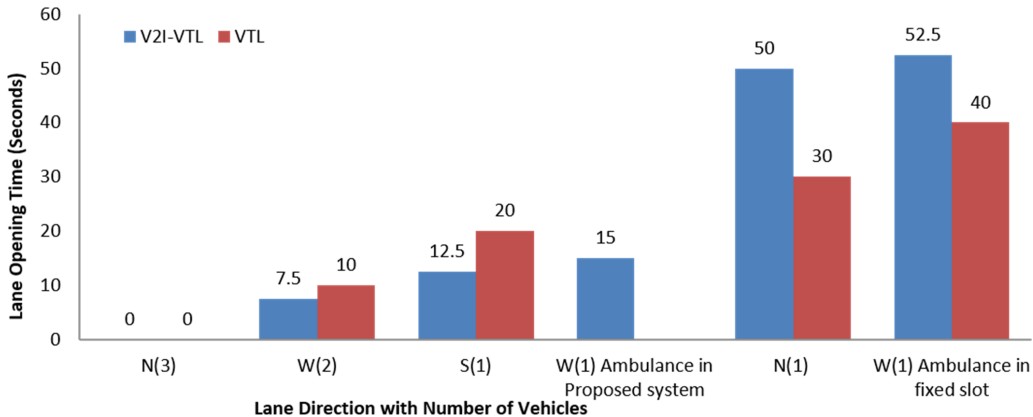

**Figure 25**  Lane opening time of emergency vehicle approaching from West.

delays for both emergency and non-emergency vehicles compared to traditional vehicle-to-traffic-light (VTL) schemes. Furthermore, by dynamically adjusting traffic light durations based on traffic density on each approach, the V2I-VTL scheme effectively mitigates congestion and minimizes vehicle queues at intersections. The successful implementation and validation of the hardware components further attest to the practical viability of the proposed scheme. Additionally, the integration of $CO_2$ emission reduction strategies underscores the environmental benefits of the V2I-VTL scheme, contributing to sustainable urban mobility solutions.

The V2I-VTL system has shown significant improvements in traffic management and emissions reduction, but future work could explore the integration of machine learning to predict congestion and dynamically adjust signals in real-time. Expanding the system to more complex road networks and integrating it with broader intelligent transportation systems (ITS) could enhance scalability and efficiency. Further research could also examine the environmental benefits in urban areas and contribute to sustainability efforts. Overall, adopting such systems could improve urban mobility, safety, and emergency response times, with broader societal impacts in creating smarter and more efficient cities.

### Funding
The authors received no funding for this work.

### Competing Interests
The authors declare there are no competing interests.

### Author Contributions
- Ajmal Khan conceived and designed the experiments, performed the computation work, prepared figures and/or tables, and approved the final draft.

- Shams ur Rahman conceived and designed the experiments, performed the computation work, prepared figures and/or tables, and approved the final draft.
- Farman Ullah performed the experiments, authored or reviewed drafts of the article, and approved the final draft.
- Muhammad Ilyas Khattak performed the experiments, authored or reviewed drafts of the article, and approved the final draft.
- Mohammed M. Bait-Suwailam analyzed the data, prepared figures and/or tables, and approved the final draft.
- Hesham El Sayed analyzed the data, authored or reviewed drafts of the article, and approved the final draft.

### Data Availability

The code is available in the Supplementary File.

### Supplemental Information

Supplemental information for this article can be found online at http://dx.doi.org/10.7717/peerj-cs.2507#supplemental-information.

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
