# Peer review of "V2I-VTL: IoT-Enabled adaptive traffic light controller and emission reduction at intersection"

_PeerJ Computer Science, doi:10.7717/peerj-cs.2507_

## Round 0.1 · original submission · Major Revisions

We have received two primarily positive reviews of your manuscript. Please, address all the issues/answer the questions raised by the reviewers.

Reviewer 1 ·

Basic reporting

Clarity and Language:
The manuscript is generally well-written, with clear and unambiguous language. The technical terms are used correctly, and the overall flow of the text is logical. However, there are several areas where the phrasing could be more concise or where minor grammatical errors need correction. Additionally, some sections would benefit from a more formal tone, particularly in the discussion and conclusions, where the language sometimes leans toward conversational. Please perform a thorough proofreading to correct minor grammatical errors and enhance the overall readability.

Literature References and Background:
The introduction and related work sections are comprehensive, providing sufficient background on the topic and citing relevant literature. However, there is room for improvement in contextualizing how this work fits into the broader field. Some references are dated, and it would be beneficial to include more recent studies to ensure the discussion is up-to-date.

Structure, Figures, and Tables:
The manuscript follows a standard structure that is easy to follow. Figures and tables are generally well-prepared, with relevant content that supports the text. However, the resolution of some figures could be higher, and their descriptions could be more detailed to ensure clarity.

Experimental design

Research Question and Knowledge Gap:
The research question is clearly defined, focusing on the development of an adaptive traffic light controller that prioritizes emergency vehicles and reduces emissions. The manuscript effectively identifies the knowledge gap it intends to fill, which is the lack of real-world validation of such systems.

Methodology Detail and Reproducibility:
The methods are described in sufficient detail to allow for replication of the study. The step-by-step explanation of the algorithms and hardware setup is thorough. However, the manuscript could provide more detailed information on the parameters used in the simulation and the specific hardware components employed.

Validity of the findings

Data Robustness and Statistical Soundness:
The data presented in the manuscript is robust and statistically sound. The results are well-supported by the data, and the use of multiple metrics (e.g., lane opening time, CO2 emissions) strengthens the findings. However, the manuscript could benefit from a more detailed statistical analysis, particularly regarding the comparison between the proposed system and existing systems.

Conclusions and Original Research Question:
The conclusions are well-stated and directly linked to the original research question. The manuscript does a good job of limiting claims to what is supported by the data. However, the conclusion could be expanded to discuss potential future work and the broader implications of the findings.

·

Basic reporting

Minor issue with citation format. In line 41, "Department (2022)" and line 43, "Administration (2021)". Are the word "Department"/"Administration" a citation source?

Experimental design

1. In chapter 4.2.4 the author mentioned CO2 emission calculation. Although the author provide an emission estimation model, to the reviewer's knowledge, VISSIM does not provide realistic individual vehicle dynamic/powertrain model, the CO2 calculation is only based on acceleration and deceleration which could be far from realistic. Additionally, the reviewer failed to find different types of vehicle are simulated, for example, light duty vs. heavy duty vs. electrical vehicles (who may have 0 emission), all could have very different emission output. The reviewer think the simulation should be redesigned, adding different types of vehicles, and considering of co-simulation: using VISSIM with other vehicle simulator (such as IPG CarMaker, Simulink, or dSPACE ASM) to get more realistic emission result.

In addition, the paper did not perform any real world emission test, which further weakened the simulation conclusion.

2. In simulation, the author assume all vehicles are travelling at a speed of 6km/h in Table 1. And in Table 2, the number are 7.2km/h. There speed are way too slow compared to real world situation. Also, the distance assumed in Table 2 (15 meter) for 7.2km/h is way to large. Can the author provide explanation how these number are chosen?

3. The paper consider using V2I-VTL to manage an intersection to let the Emergency Vehicles have higher priorities. However, in real-world scenario there might be 3 EVs coming from North, East and South to a four way intersection, and all trying to go West bound. This paper failed to mention how this type of scenario is handled. It is suggest to discuss scenario where multiple EVs approaching from different directions.

Validity of the findings

no comments

---

## Round 0.2 · accepted · Accept

The authors have addressed all the issues raised by the reviewers.

Reviewer 1 ·

Basic reporting

The revised manuscript is well-written with clear English. The literature review is comprehensive, providing sufficient context. The article is structured appropriately, and figures and tables are generally well-prepared.

Experimental design

The research question in the revised manuscript is well-defined and addresses a relevant gap in the field. The experimental design is robust, with detailed methods that ensure replicability.

Validity of the findings

The findings in the revised manuscript are valid, supported by robust data and sound statistical analysis. The conclusions are clearly linked to the research question and supported by the results.

Additional comments

Overall, the revised manuscript is of high quality, with significant contributions to the field. It is recommended for acceptance.

·

Basic reporting

See Additional comments

Experimental design

Additional comments

Validity of the findings

Additional comments

Additional comments

About the "Distance assumed" in Table 3 (15 meter), the reviewer original thought it was the stopping distance between each vehicle, that's why I think "15 meter for 7.2km/h is way to large". The author has explained that this is the vehicle distance from the intersection, that addressed my concern.

The author has address all of my comments, great work!